# Real-time Reconstruction of Human Visual Perception from fMRI

## Abstract

Closed-loop neurofeedback based on functional magnetic resonance imaging (fMRI) is an emerging methodology that has led to important scientific and clinical advances. However, the sophistication of the analysis methods used in real-time fMRI lags behind the state-of-the-art in fMRI decoding, largely due to computational factors: Most scanning facilities lack the hardware and software infrastructure to implement these state-of-the-art pipelines in real-time. A further challenge is adapting such pipelines to fit within the computational envelope of real-time processing, where the analysis needs to be conducted in a matter of seconds and without leveraging data acquired later in the session. Here, using RT-Cloud, an open-source, scalable cloud-based platform for real-time fMRI, we demonstrate that it is possible to implement a state-of-the-art algorithm for reconstructing perceived natural images (Mind-Eye2). This work is an initial step towards deploying these powerful fMRI decoding pipelines to closed-loop settings, paving the way for their use in non-invasive brain-computer interfaces for scientific discovery and clinical treatment.

## 1. Introduction

A longstanding goal for cognitive neuroscience is to develop non-invasive methods for decoding thoughts based on brain activity – the finer-grained the decoding, the more useful this information will be for downstream scientific and clinical applications. Of all the non-invasive brain imaging methods, functional magnetic resonance imaging (fMRI) holds the greatest promise of enabling such fine-grained decoding of cognitive states, given its superior spatial resolution compared to other non-invasive methods like electroencephalography (EEG) and magnetoencephalography

(MEG). Indeed, in recent years there have been rapid advances in our ability to decode fine-grained sensory information from fMRI, largely driven by the advent of foundation models in computer vision (Radford et al., 2021) and natural language processing (Radford et al., 2019; Devlin et al., 2019; Touvron et al., 2023). These advances have resulted in impressive feats of mind-reading, for example by reconstructing the images that a participant has seen (Ozcelik et al., 2022; Ozcelik & VanRullen, 2023; Scotti et al., 2023; 2024) or imagined (Kneeland et al., 2025b), or the contents of a narrative that they listened to (Tang et al., 2023).

In parallel, there have also been advances in deploying fMRI in real-time settings, e.g., in closed-loop neurofeedback paradigms where participants are guided towards targeted brain states by feedback based on BOLD (Blood Oxygen Level-Dependent) signals that are processed on-the-fly. These studies have provided important insights into clinical conditions such as depression (Young et al., 2017; Mennen et al., 2021) and have also shed light on fundamental learning mechanisms in the brain (Peng et al., 2024). However, in comparing recent real-time fMRI studies to state-of-the-art methods used for decoding in offline analyses, there is a clear disconnect. The multivariate analysis methods used in recent real-time studies (e.g., decoding whether people are attending to a scene or a face) are sophisticated compared to prior methods in fMRI neurofeedback (e.g., tracking the average activation of the amygdala), but they are nowhere near as sophisticated as state-of-the-art fMRI decoding methods that enable fine-grained reconstruction of individual stimuli. Importing such sophisticated decoding methods into real-time fMRI pipelines would vastly improve their utility as brain-computer interfaces, enabling novel neurofeedback experimental designs (e.g., allowing patients with depression to see how their perception of an image overemphasizes negative features of the image). The main obstacle to doing this is computational. State-of-the-art fMRI decoding methods are computationally intensive, relying on software and hardware infrastructure that most scanning facilities lack. Finally, it is challenging to adapt these methods to the inherent structural constraints of real-time fMRI: conducting the analysis within seconds, without access to the data collected later in the session.

In the present work, we chose reconstruction of seen images as the ideal use-case to demonstrate the feasibility of adapt-

[1]Anonymous Institution, Anonymous City, Anonymous Region, Anonymous Country. Correspondence to: Anonymous Author <anon.email@domain.com>.

Preliminary work. Under review by the International Conference on Machine Learning (ICML). Do not distribute.

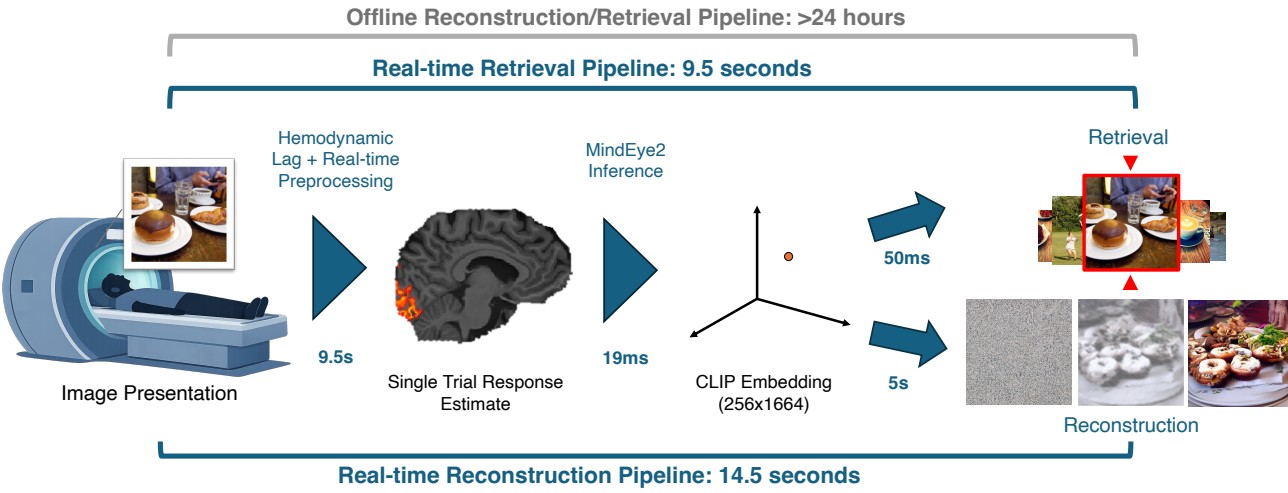

*Figure 1.* Real-time fMRI-to-image decoding pipeline. The participant views the image in the scanner; the measured BOLD signal undergoes real-time compatible preprocessing (motion correction and registration) and is deconvolved to extract a single-trial response estimate (beta). MindEye2 inference maps betas to CLIP latent embeddings, which can then be used to reconstruct the seen image or retrieve the image out of a pool of candidates within seconds after it is first viewed.

ing a computationally heavy machine learning workload for real-time fMRI. In doing so, we overcame prior challenges to achieve the first real-time reconstruction of visual perception from fMRI. To accomplish this, we use RT-Cloud (Wallace et al., 2022), a scalable open-source framework that enables real-time streaming and analysis of fMRI data, in conjunction with MindEye2 (Scotti et al., 2024), a large (>700M parameters) machine learning architecture for visual decoding. We conducted tests with a 3 Tesla (3T) MRI scanner, a platform available at tens of thousands of facilities at hospitals and universities worldwide, rather than the less prevalent 7 Tesla (7T) platform that supported much prior work in this space. We demonstrate that, by fine-tuning our model with just one hour of data from a new participant, our pipeline can support fine-grained visual decoding in a subsequent session, reconstructing images within seconds after they are viewed by the participant.

After developing these methods in simulation using previously collected 3T data, we successfully implemented this in a real-time fMRI session where we displayed reconstructions within seconds after the participant viewed the images. Here, we present simulated real-time analyses to test variations in preprocessing and analysis.

The main contributions of this work are as follows:

- We demonstrate for the first time that it is possible to reconstruct seen images from fMRI at single-trial resolution, in real-time (as fast as 15 seconds post image onset), improving on prior approaches that span days.

- We show that reliable real-time decoding is achievable with minimal training data (~1 hour) from a new partici-

pant and can be extended from 7T to 3T fMRI.

- We compare several real-time setups that trade off decoding performance with latency, quantifying the drops in performance one can expect transitioning from offline to real-time analysis.

- We demonstrate that the RT-Cloud software framework can support cutting-edge machine-learning-heavy analysis pipelines.

- We make all code, models, and data publicly available (links removed for anonymous submission) and carry out development fully in the open to encourage collaboration and downstream applications.

## 2. Methods

In this section, we describe step-wise modifications to our pipeline, starting with offline analysis (Scotti et al., 2024) and working our way down to the fastest-possible real-time analysis (fMRI has intrinsic temporal limitations; see Section 2.3.2). This serves to document the progressive trade-off between processing time and performance.

Training and evaluation are kept constant across pipeline variations; this facilitates fair comparisons and enables us to attribute differences in performance to the components of the pipelines. To this end, all of the results shown below involve pretraining on data from 7 participants from the Natural Scenes Dataset (NSD; Allen et al., 2022); we fine-tune on only one session of data (~1 hour) from a new participant we collected with 3T fMRI, and we evaluate on a subsequent session from the same participant. We additionally replicate

key findings using the held-out NSD subject. Overall, these results highlight how we are able to achieve above-chance real-time decoding for a new participant within just two scanning sessions.

## 2.1. 3T fMRI Data Collection

### 2.1.1. PARTICIPANT

The 3T fMRI data were collected from an author of this study (initials, age, and sex removed for anonymous submission). The subject was healthy, had normal vision, and was right-handed. Written informed consent was obtained from the participant and the experimental protocol was approved by the university's Institutional Review Board (name of university removed for anonymous submission). The participant was compensated at a standard rate of $30 per hour, plus bonuses for return visits to the scanner.

### 2.1.2. STIMULI

The stimuli used in this study were natural images (including scenes, animals, and people), most of which were from NSD, which itself sampled from Microsoft COCO (Lin et al., 2015).

Both scan sessions (training and real-time) involved viewing 531 unique images. This included 50 images from the "special515" subset of NSD images, each presented three times; 419 additional images from NSD, each presented once; and 31 pairs of highly similar images (e.g., two lighthouses), where each of these 62 items were presented twice. The paired images (some from Wanjia et al., 2021 and some generated using StyleGAN from Karras et al., 2019) were included to support future investigations of the model's ability to distinguish fine-grained representations but were not analyzed further here (although they were included in the training data for the model). Counting repetitions, this totaled 693 image presentations per scan.

### 2.1.3. EXPERIMENTAL DESIGN

We employed a rapid event-related design closely matched to NSD. Each session consisted of 11 functional runs of 63 images presented every 4 seconds. Each image appeared for 3 seconds with a 1-second inter-stimulus interval. The order of presentation was pseudo-random with a constraint to prevent back-to-back repeated presentations of the same image. Blank trials were interspersed to assist HRF (hemodynamic response function) deconvolution (see Appendix 4.1.2).

The subject was instructed to fixate on a small red dot that was present throughout the experiment, and to avoid eye movements even if drawn to look at details in different parts of an image. When an image was presented, the subject's task was to determine whether the image was "new" or "old", responding with a press of "1" or "2" on a button-box,

respectively. At the end of each run, they received accuracy feedback. This familiarity detection task served as a way to maintain the subject's attention (as in Allen et al., 2022) and responses were not analyzed.

## 2.2. 3T fMRI Acquisition

MRI data were collected with a 3T Siemens Prisma scanner with a 64-channel head coil (institution removed for anonymous submission). Functional scans were acquired using a T2*-weighted multiband EPI sequence (repetition time [TR] = 1500 msec, echo time [TE] = 33 msec, voxel size = 2.0 mm isotropic, flip angle = 70°, multiband factor = 4, 52 slices automatically aligned to the AC-PC line). These slices comprised a partial volume fully covering the occipital and temporal lobes. Additional details in Appendix A.7.

## 2.3. 3T fMRI Data Preprocessing

In both the offline and real-time preprocessing pipelines, we analyzed the subject's data in native anatomical T1w-space.

### 2.3.1. OFFLINE PREPROCESSING

All offline preprocessing was performed using fMRIPrep (version 24.0.1; Esteban et al., 2019). Single-trial responses to images ("betas") were estimated using GLMsingle (Prince et al., 2022). Repeated image presentations are used by GLMsingle to improve single-trial response estimates. Repetitions are also used to determine reliable voxels, as described in Section 2.6.2. Betas were z-scored voxelwise using the training images from the entire session.

### 2.3.2. REAL-TIME PREPROCESSING

Each TR, a 3-D brain volume is streamed from the MRI scanner. Once, at the beginning of the session, we compute a linear (affine) registration between the first functional volume and a BOLD reference volume from the offline training session using FSL's (version 6.07.15; Jenkinson et al., 2012) FLIRT tool (version 6.0) with 6 degrees of freedom. On all subsequent TRs, we use FSL's MCFLIRT to compute a motion correction transformation for the current volume with reference to the first functional volume of the session. The two transformation matrices (for registration and motion correction) are then combined and applied to the data from the current TR to align the brain volume to the reference data used for fine-tuning.

One challenge with real-time fMRI is that, when a participant views an image, the BOLD response is temporally extended, generally peaking ∼4-6 seconds after stimulus onset and not fully dissipating until ∼12–20 seconds. To address this, we delay inference until ∼7.9 seconds (∼2 trials) after the onset of the target image to increase the likelihood of capturing the BOLD response peak (see Appendix A.8

for additional details). The implications of this intrinsic temporal limitation of fMRI are discussed further in Section 4.2. The second challenge is that, due to this lag, the response to a given image may overlap with that of the next image (if, as here, they are presented in rapid succession). To address the second challenge, we take a standard approach in fMRI data analysis, fitting a GLM (general linear model) to deconvolve the temporally-extended response to an image and extract a single-trial beta. Here, we use a canonical HRF (Nilearn version 0.10.4; Contributors; Abraham et al., 2014; Pedregosa et al., 2011) to model the BOLD response. Voxelwise betas are z-scored cumulatively as the session progresses.

## 2.4. Pipeline Variations

The temporal resolution of fMRI ($\sim$0.5-1 Hz) and the extended BOLD response (spanning $\sim$6-12 seconds) place an inherent lower bound on real-time decoding speed. As a result, we document results using different stimulus delays. Specifically, the "delays" refer to the amount of time we allow to elapse before attempting to fit the GLM (see Appendix A.8 for additional details).

Here, we describe three main analysis speeds that fall under the umbrella of real-time compatible analysis. We define "real-time compatible" as an analysis that can be used to influence the experiment while the subject is still being scanned (e.g., adapting the stimuli shown to the subject based on some readout of brain activity or providing neurofeedback to modulate activity). We call the three variations "fast", "slow", and "end-of-run". For "fast" real-time, we acquire functional data for $\sim$7.9 seconds ($\sim$2 trials) post stimulus-onset (Table 2), which should be sufficient for the BOLD response for most voxels to peak. Preliminary explorations demonstrated that a lower amount of time is insufficient for decoding. For "slow" real-time, we acquire functional data for $\sim$29 seconds ($\sim$7 trials) post stimulus-onset, an intermediate point that optimizes the trade-off between delay and performance (Section 3.2). Finally, for "end-of-run" real-time, we acquire functional data until the end of the functional run – which lasts roughly 5 minutes – before fitting the GLM for each trial.

## 2.5. Model Architecture

We used a condensed version of the MindEye2 architecture (Scotti et al., 2024), omitting the low-level submodule, image-to-image refinement, and text caption refinement steps in addition to reducing the shared-subject latent dimensionality from 4096 to 1024. The authors of the original work reported that these changes minimally impacted evaluation metrics, but their removal lowers the parameter count and significantly speeds up training and inference.

The inputs (voxelwise single-trial betas within a "reliability mask"; see Section 2.6.2) are converted to a shared-subject latent space using ridge regression (Hastie et al., 2009). These latents are passed through a residual multilayer perceptron (MLP) and a linear layer, resulting in an embedding with the same dimensionality as OpenCLIP ViT-bigG/14's image token embeddings (penultimate layer; $256 \times 1664$ dimensions) (Radford et al., 2021). This "backbone" embedding is then passed to two separate branches: image reconstruction (generate an image) and image retrieval (select the seen image out of a pool of candidates). For reconstruction, the backbone embedding is mapped to the CLIP model's image embedding space guided by a diffusion prior, and a frozen unCLIP model based on Stable Diffusion XL (Podell et al., 2023) then generates the image reconstruction. For retrieval, the backbone embedding follows an analogous mapping to CLIP image space – but via an MLP "projector" rather than a diffusion prior. The resulting (predicted) fMRI-CLIP embedding can then be compared with the (ground-truth) CLIP image embeddings of candidate images; the top-$k$ retrievals are defined as the $k$-nearest neighbors (based on cosine similarity) to the predicted embedding. See Figure 1 for an overview of the real-time analysis pipeline.

## 2.6. Model Training

Prior to inference (offline or real-time), we follow the training procedure described by Scotti et al. (2024). We use a pretrained checkpoint from this paper trained on 7 subjects from NSD (Allen et al., 2022). Subsequently, we fine-tune the model on one session ($\sim$1 hour) of 3T data from a new participant. Importantly, we did not perform any model training or hyperparameter selection based on the (held-out) session used for evaluations.

### 2.6.1. TRAIN AND TEST SPLIT

During pretraining, the model was trained on all images except the "shared1000", which were seen by all NSD participants ("special515" refers to a subset of shared1000 for which all participants saw all three repetitions). For the 3T participant, the model was fine-tuned on all 543 images (481 unique) from the first scan except the 50 "special515" images (repeated 3x each; 150 total). The model was then tested on three repeats of a different subset of 50 "special515" images from the second (real-time) scan session. During pretraining and fine-tuning on 7T data, the inputs to the model were single trial betas, while betas averaged across three repetitions were used for fine-tuning for 3T.

### 2.6.2. SELECTING RELIABLE VOXELS

First, we transformed the "nsdgeneral" region of interest (ROI), a mask of voxels in occipital cortex that responded reliably in NSD, from the template MNI space into the

subject's native anatomical T1w space. We then applied a subject-specific "reliability mask" on top of the nsdgeneral mask. Reliability for a given voxel is defined as the average correlation of responses for repeated presentations of the same image. Specifically, we computed the Pearson's correlation of betas (from GLMsingle) across the first two instances of each repeated image in the training session and set a reliability threshold at $r > 0.2$ to generate a binary mask. We found that the addition of a subject-specific reliability mask improved performance over the nsdgeneral ROI in some preliminary data.

### 2.7. Model Evaluation

The test set for all experiments (50 "special515" images, repeated 3x each) was kept identical to facilitate comparisons across variations of the analysis. Unless otherwise stated, evaluations use single-trial betas from the first presentation of the three repeats only.

PixCorr (pixel correlation), SSIM (structural similarity index metric; Zhou Wang et al., 2004), EfficientNet-B1 ("Eff"; Tan & Le, 2020) and SwAV-ResNet50 ("SwAV"; Caron et al., 2021) compute the average correlation distance between the ground truth and reconstructed image. AlexNet (layers 2 and 5) (Krizhevsky et al., 2012), Inception-v3 (last pooling layer) (Szegedy et al., 2016), and CLIP (last layer of ViT-L/14) (Radford et al., 2021) use two-way comparisons based on extracted features from the specified layer (chance=50%). For each test image, we compute the Pearson's correlation of each ground truth image with each reconstruction in the test set; two-way accuracy refers to the number of times the correct ground-truth-reconstruction pair is more correlated than a mismatched pair, averaged over all possible pairwise comparisons and then over all test images to produce a single score.

We include two retrieval metrics: "image" retrieval and "brain" retrieval. Image retrieval uses the predicted CLIP embedding (based on fMRI) to choose the nearest neighbor ground-truth CLIP embedding. Brain retrieval is the opposite, using a ground-truth CLIP embedding to choose the nearest neighbor fMRI-CLIP embedding. The retrieval score in each case is the top-1 accuracy when repeating this procedure for each image in the test set (chance=2%; 1/50).

## 3. Results

### 3.1. Comparing Pipeline Variations

The performance of the offline 3T pipeline is close to offline 7T performance (Table 1, Figure 3). Previous work relied on 7T MRI; our results show that good offline fMRI-to-image performance is possible using a model that is pretrained on 7T data and then fine-tuned and tested on 3T data (Figure 2; see Figure 10 for more examples). Note that the 7T and 3T

tests were conducted on different participants, so we caution against attributing the difference in performance exclusively to scanner field strength.

As expected, fully offline analyses perform better than their real-time compatible counterparts across most evaluation metrics (Table 1, Figure 3). The performance gap between the offline and end-of-run pipelines suggests that the inclusion of extensive preprocessing steps such as fMRIPrep and GLMsingle is beneficial, but not essential, for single-trial decoding. Importantly, even the fastest real-time analysis performs above chance-level (even without pretraining on NSD; Appendix A.1). We qualitatively replicate these results using the held-out NSD subj01 (Appendix A.2).

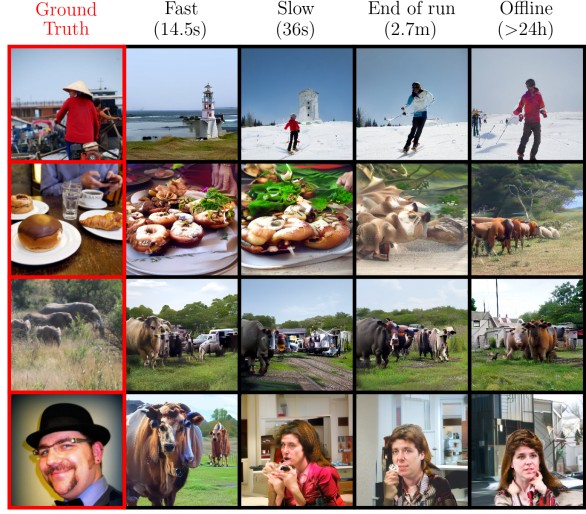

*Figure 2.* Hand-picked example reconstructions for different configurations in 3T.

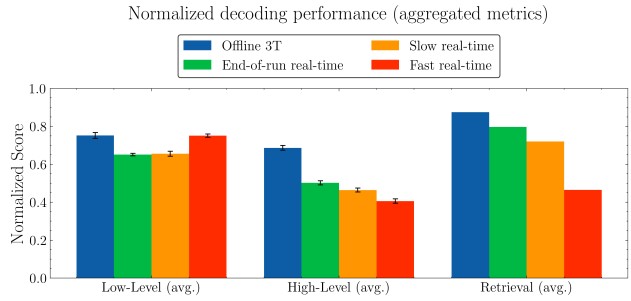

*Figure 3.* Aggregated evaluation metrics for 3T offline and real-time pipelines. Scores are min-max normalized per metric so that 0 corresponds to using random COCO images as reconstructions and 1 corresponds to offline NSD performance. Bars show the mean and standard error across 5 random seeds.

### 3.2. Stimulus Delay vs. Performance

There is a positive relationship between stimulus delay (i.e., how long we wait before starting to analyze the response to a stimulus) and decoding performance, implying that

| Method | Latency | Low-Level | | | | High-Level | | | | Retrieval | |
|---|---|---|---|---|---|---|---|---|---|---|---|
| | | PixCorr ↑ | SSIM ↑ | Alex(2) ↑ | Alex(5) ↑ | Incep ↑ | CLIP ↑ | Eff ↓ | SwAV ↓ | Image ↑ | Brain ↑ |
| Offline NSD (avg. 3 reps.) | 1d | 0.231 | 0.349 | 88.4% | 95.6% | 85.9% | 78.8% | 0.803 | 0.423 | 100% | 96% |
| Offline 3T (avg. 3 reps.) | 1d | 0.16 | 0.343 | 86.6% | 91.1% | 75.9% | 75.4% | 0.854 | 0.493 | 90% | 88% |
| Offline NSD | 1d | 0.228 | 0.330 | 84.5% | 93.1% | 85.5% | 78.5% | 0.832 | 0.448 | 78% | 82% |
| Offline 3T | 1d | 0.101 | 0.328 | 79.7% | 83.8% | 73.2% | 72.1% | 0.884 | 0.516 | 76% | 64% |
| End-of-run real-time | 2.7m | 0.052 | 0.330 | 75.4% | 79.8% | 68.9% | 65.0% | 0.918 | 0.544 | 66% | 62% |
| Slow real-time | 36s | 0.062 | 0.334 | 73.8% | 77.3% | 68.6% | 63.7% | 0.928 | 0.550 | 58% | 58% |
| Fast real-time | 14.5s | 0.088 | 0.350 | 74.0% | 75.6% | 64.6% | 62.9% | 0.926 | 0.572 | 36% | 40% |

*Table 1.* Latency and reconstruction/retrieval metrics for 3T and 7T offline and real-time pipelines. Reconstruction metrics are averaged over 5 random seeds; retrieval is deterministic (see Section 2.7 for details on computing metrics). ↑ (↓) means higher (lower) scores are better.

collecting additional data before fitting the GLM improves the quality of the response estimates (Figure 4). This relationship peaks early with an elbow at roughly 30 seconds before diminishing returns, which suggests that this may be an optimal trade-off point between speed and accuracy. We were able to qualitatively replicate this pattern of findings when we implemented our real-time compatible pipeline on NSD subj01 (Appendix A.3).

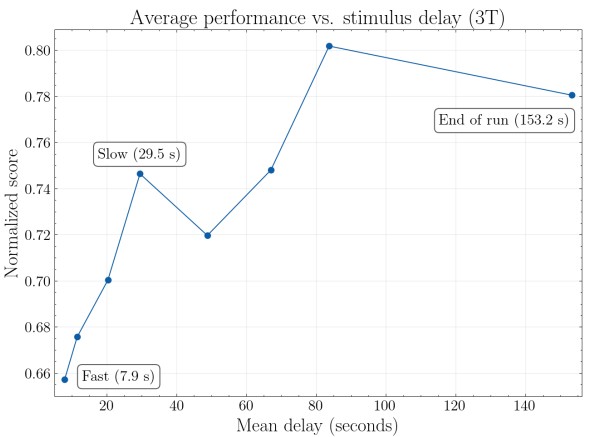

*Figure 4.* Average normalized performance across stimulus delays in 3T. Metrics are min-max normalized per metric using all models and delays, then averaged across metrics. Scores are then linearly rescaled per metric so that 0 corresponds to using random COCO images as reconstructions and 1 corresponds to offline NSD performance. Stimulus delay indicates the time elapsed after stimulus presentation before starting to analyze the neural response.

### 3.3. Real-time Analysis Latency

The amount of time needed for motion correction, registration, MindEye inference, reconstruction, and retrieval is the same for all real-time processing variants (Table 2). This is because the same operations are being executed regardless of the stimulus delay, and the input data are always the same size. Fitting the GLM is slightly slower for longer delays because there is more functional data present per trial

as the delay grows. It takes roughly 7 seconds to perform

| | Fast (s) | Slow (s) | End-of-run (s) |
|---|---|---|---|
| *Stimulus Delay* | $7.85_{\pm 0.59}$ | $29.45_{\pm 2.63}$ | $153.18_{\pm 79.63}$ |
| Motion Correction | $0.39_{\pm 0.01}$ | $0.39_{\pm 0.01}$ | $0.39_{\pm 0.01}$ |
| Registration | $0.18_{\pm 0.00}$ | $0.18_{\pm 0.00}$ | $0.18_{\pm 0.00}$ |
| *GLM Fit* | $1.09_{\pm 0.12}$ | $1.11_{\pm 0.11}$ | $1.21_{\pm 0.08}$ |
| Inference | $0.19_{\pm 0.01}$ | $0.19_{\pm 0.01}$ | $0.19_{\pm 0.01}$ |
| Reconstruction | $4.56_{\pm 0.05}$ | $4.56_{\pm 0.05}$ | $4.56_{\pm 0.05}$ |
| Retrieval | $0.50_{\pm 0.00}$ | $0.50_{\pm 0.00}$ | $0.50_{\pm 0.00}$ |
| Total Latency | $14.76_{\pm 0.62}$ | $36.38_{\pm 2.65}$ | $160.21_{\pm 79.65}$ |

*Table 2.* Time for real-time fMRI preprocessing and MindEye2 inference (mean ± standard deviation over all session trials). Steps where processing time differs across variations are *italicized*.

fMRI preprocessing and inference in total. After adding this computational lag to the pre-defined stimulus delay (the difference between stimulus onset and the start of analysis), the "fast" real-time pipeline takes on average ∼14.8s per trial from stimulus onset to reconstruction and retrieval. The "slow" pipeline takes ∼36s and "end-of-run" ∼160s.

## 4. Discussion

### 4.1. Real-time vs. Offline Performance

Our results reveal that averaging representations over multiple repetitions of the same image and acquiring more data before estimating a single-trial response are the two most important factors supporting decoding.

#### 4.1.1. AVERAGING ACROSS REPETITIONS

Table 1 demonstrates the large impact of averaging repetitions on evaluation metrics. We reported results from only the first presentation of each test image in real-time pipelines, but averaging over stimulus repetitions is standard in neuroimaging analysis because this maximizes signal-to-noise (SNR) of the data, increasing the likelihood that the decoded representation is closer to the true underlying brain response to a given stimulus. This is important for fMRI

and other non-invasive methods, which have relatively low SNR compared with their invasive counterparts. Indeed, related work using electroencephalography (EEG, Kneeland et al., 2025a) and magnetoencephalography (MEG, Benchetrit et al., 2024), which have better temporal resolution but worse spatial resolution than fMRI, rely on 12 and even up to 80 repetitions of the same image to extract a reliable signal.

### 4.1.2. STIMULUS DELAY

Depending on the use-case, a researcher may wish to select a particular point along the trade-off continuum between real-time decoding speed and accuracy. For instance, closed-loop neurofeedback aiming to modulate the cortical representations of individual stimuli would require single-trial decoding in the least time possible. On the other hand, an investigation of emotional processing for patients with a mood disorder may want to aggregate stimuli across several minutes before delivering an intervention (e.g., personalized transcranial magnetic stimulation parameters based on a cognitive signature). Our real-time stimulus delay comparisons shed light on the relative performance one might expect to trade-off across this continuum.

Given the temporally extended nature of the BOLD response, increasing the time window prior to deconvolving the HRF should allow for better response estimates. Since the BOLD signal is known to decay to baseline on the order of 20 seconds ($\sim$5 trials), one might expect that increasing stimulus delay further would not affect decoding performance. Surprisingly, we observe that performance continues improving as the delay increases to include data from the entire functional run. We further replicated this using the held-out NSD subject (Appendix A.3).

We offer two (non-mutually exclusive) explanations. First, the GLM approach to HRF deconvolution models the hemodynamic response of the trial of interest, but not in isolation; other trials (and thus images) are implicitly modeled as eliciting zero response from a given voxel. The betas estimate the response of a voxel to an image *assuming that it does not respond to any other image*, so including more images may be beneficial as the GLM is able to gain information related to a voxel's *differential* selectivity for an image, which the machine learning architecture may learn to take advantage of. Secondly, the inclusion of blank trials in the design allows the BOLD response to decay; without these trials, a voxel that responds to all (or many) images may have a chronically sustained BOLD response. This is due to the rapid presentation of images (here every 4 seconds) which may result in additive interactions between responses to neighboring images. Including more data in the GLM may improve response estimates by resolving these ambiguities.

### 4.2. Limitations

Functional MRI has relatively poor temporal resolution compared with other non-invasive methods (EEG and MEG) and invasive methods, which can resolve neural activity on the order of milliseconds. This inherent temporal limitation places a lower bound on closed-loop feedback and brain-computer interface applications, though fMRI remains the most spatially resolved non-invasive brain imaging tool, motivating its use here. FMRI is also not portable, requiring participants to visit an MRI facility (generally located at a university or a hospital). Additionally, repeated or long-term interventions are difficult, requiring multiple visits to the scanner; on the other hand, invasive brain-computer interfaces are implanted once (often with neurosurgery) and can subsequently record data on a continuous basis for the lifespan of the device (Pels et al., 2019; Davis et al., 2025; Mitchell et al., 2023).

One limitation of our 7T pretraining procedure is that the images used for pretraining were interleaved with some of the images that were later (in a separate session) used for testing. Due to the temporal lag of the BOLD response, this could have led to a minor form of data leakage, whereby the neural response to the test images affects the beta maps for images presented after them during pretraining; this issue could be addressed in future work by defining a train/test split during pretraining based on continuous blocks of time as in Careil et al. (2025). Importantly, we expect that this issue would not significantly inflate our results, given that the model was only exposed to the fMRI data from the test images, but not the labels (CLIP embeddings) for these images. Our use of a fully held-out session for evaluation further minimizes this risk.

### 4.3. Future Directions

#### 4.3.1. ADDITIONAL TRAINING DATA

Banville et al. (2025) have demonstrated that collecting additional data per subject is the primary driver of image decoding performance (as opposed to increasing the total number of subjects). In line with these findings, we demonstrate improved real-time decoding performance with one additional fine-tuning session in 3T and replicate this using more sessions from NSD (Appendix A.4). Future work may explore this relationship further, as it is possible that some researchers may have the bandwidth to perform multiple training sessions prior to real-time inference in order to maximize decoding performance.

#### 4.3.2. IMPROVED STATE-OF-THE-ART AND SCALABLE METHODS

Future analysis pipelines and machine learning architectures may supersede the ones used here as more effective meth-

ods are developed. For example, Careil et al. (2025) have demonstrated a method for time-resolved decoding from fMRI and Beliy et al. (2025) have shown better transfer learning in limited data settings. Improved pipelines such as these and others that are developed in the future can easily replace the core MindEye2 decoding architecture used here within the RT-Cloud framework for real-time fMRI data streaming.

In addition, the field may increasingly benefit from the incorporation of foundation models: self-supervised pretraining approaches leveraging large quantities of unlabeled data. For example, recent work has demonstrated impressive transfer after pretraining, even generalizing across tasks and species (Azabou et al., 2023; Zhang et al., 2025; Ryoo et al., 2025) with a novel tokenization approach. These self-supervised pretraining methods have already proven highly impactful in deep learning; the intuition for brain foundation models as a backbone for decoding is that these architectures can take advantage of large-scale unlabeled data to implicitly learn useful domain-general latent representations and inductive biases beyond what can be learned from smaller, task-specific labeled datasets (Lane et al., 2025). Using these powerful latent representations, foundation models can then be fine-tuned to a specific task using a smaller amount of labeled data, improving performance. Such approaches may further extend the benefits of cross-subject pretraining demonstrated in Scotti et al. (2024) that we leverage here, potentially reducing the amount of data required from a new participant and increasing the ceiling on decoding performance.

In the coming years, the most effective approaches to fMRI decoding are likely to be large machine learning architectures comprising millions or even billions of parameters. The real-time fMRI platform, RT-Cloud, can support these approaches by enabling real-time inference using scalable cloud computing.

### 4.4. Towards Non-Invasive Brain-Computer Interfaces

Although the reconstructions that our real-time pipelines produce are not as impressive as the state-of-the-art given unlimited processing time, they are clearly above chance both quantitatively and subjectively, generally containing some low-level visual similarities and some semantic relevance to the ground-truth image. We expect that future work will improve the quality of single-trial real-time reconstructions and close the gap between real-time and offline processing both in fMRI and beyond. This proof-of-concept demonstration should empower the field to think about how this approach can enable novel neurofeedback and brain-computer interface paradigms that were previously considered computationally infeasible. To illustrate this, we offer an outline below for possible avenues of research that build on this work.

The core aim of this family of generative approaches to fMRI-to-image reconstruction (Ozcelik et al., 2022; Ozcelik & VanRullen, 2023; Scotti et al., 2023; 2024; Beliy et al., 2025) is to learn a mapping from fMRI brain activity to the latent embedding space of a pretrained foundation model. In our case, we use CLIP (Radford et al., 2021), which is jointly trained on large-scale language and image data resulting in a rich, semantically-relevant latent space. Mapping brain activity into this latent feature space allows us to reconstruct seen images, but we wish to emphasize that this is just one specific application. By applying this mapping between brain activity and this latent embedding space in real-time, we are able to gain a dynamic, semantically-aligned readout of the participant's internal cognitive representations.

Going forward, our real-time visual decoding pipeline can potentially support a range of novel applications enabled by the decoded latent visual representations, beyond just image reconstruction. By providing closed-loop neurofeedback based on a participant's latent representation of a particular image, one can modify someone's perception of that image within the constraints of volitional control of neural activity (Motiwala et al., 2026) to perform non-invasive, causal manipulations of brain activity in humans.

For example, the stimulus-level granularity of these visual representations can facilitate neurofeedback studies that manipulate the latent representations of similar images to become more distinct from another, which could support brain-computer interface technologies to accelerate learning of fine-grained information. This kind of fine-grained neurofeedback could be applied to patients with depression, who have a known attentional bias towards negatively-valenced information (Mennen et al., 2019); using the methods described here, the patients could be shown the ground-truth image alongside a reconstruction of their negatively-biased perception of the image, thereby helping them to notice and correct for these biases.

## 5. Conclusion

We demonstrate here that it is possible to leverage a large, highly curated dataset collected using 7T fMRI (Allen et al., 2022) and fine-tune on just one session of data from a new participant in 3T fMRI to enable real-time visual decoding as early as their second visit. Beyond the proof-of-concept image decoding use case demonstrated here, similar preprocessing pipelines combined with RT-Cloud can support any real-time fMRI workflow involving complex forms of analysis. In this spirit, we encourage future work that employs such approaches to implement novel closed-loop experiments and develop non-invasive brain-computer interface technologies.

## Open Science

Removed for anonymous submission.

## Software and Data

Removed for anonymous submission.

## Impact Statement

Here, we demonstrate that it is possible to decode visual perception from fMRI as early as a participant's second visit, within seconds after they view an image. Such fine-grained decoding has long been a goal of the field, and opens the door to several novel lines of research including closed-loop experimental paradigms and brain-computer interface technologies. These may have applications in scientific discovery as well as in clinical treatments – such as by modulating the neural activity of a patient with depression towards a more normative state. The development of non-invasive brain-computer interfaces promises a window into people's ongoing cognitive processes as they unfold – which has wide applications, for example enabling communication with locked-in patients or accessing the contents of a dream in real-time.

As the field continues to develop methods in pursuit of these goals, we emphasize the importance of ethical research practice, in line with the principles of Respect for Persons, Beneficence, and Justice (National Commission for the Protection of Human Subjects of Biomedical and Behavioral Research, 1979), to which researchers have a professional, ethical, and moral obligation.

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

Contributors, N. Nilearn.

Davis, K. C., Wyse-Sookoo, K. R., Raza, F., Meschede-Krasa, B., Prins, N. W., Fisher, L., Brown, E. N., Cajigas, I., Ivan, M. E., Jagid, J. R., and Prasad, A. 5-Year Follow-up of a Fully Implanted Brain-Computer Interface in A Spinal Cord Injury Patient. *Journal of neural engineering*, 22(2):10.1088/1741–2552/adc48c, April 2025. ISSN 1741-2560. doi: 10.1088/1741-2552/adc48c.

Devlin, J., Chang, M.-W., Lee, K., and Toutanova, K. BERT: Pre-training of Deep Bidirectional Transformers for Language Understanding, May 2019.

Dosovitskiy, A., Beyer, L., Kolesnikov, A., Weissenborn, D., Zhai, X., Unterthiner, T., Dehghani, M., Minderer, M., Heigold, G., Gelly, S., Uszkoreit, J., and Houlsby, N. An Image is Worth 16x16 Words: Transformers for Image Recognition at Scale, June 2021.

Esteban, O., Markiewicz, C. J., Blair, R. W., Moodie, C. A., Isik, A. I., Erramuzpe, A., Kent, J. D., Goncalves, M., DuPre, E., Snyder, M., Oya, H., Ghosh, S. S., Wright, J., Durnez, J., Poldrack, R. A., and Gorgolewski, K. J. fMRIPrep: A robust preprocessing pipeline for functional MRI. *Nature Methods*, 16(1):111–116, January 2019. ISSN 1548-7105. doi: 10.1038/s41592-018-0235-4.

Hastie, T., Tibshirani, R., and Friedman, J. Linear Methods for Regression. In Hastie, T., Tibshirani, R., and Friedman, J. (eds.), *The Elements of Statistical Learning: Data Mining, Inference, and Prediction*, pp. 43–99. Springer, New York, NY, 2009. ISBN 978-0-387-84858-7. doi: 10.1007/978-0-387-84858-7_3.

Huettel, S. A., Song, A. W., and McCarthy, G. *Functional Magnetic Resonance Imaging*. Sinauer Associates, Sunderland, Mass, 2004. ISBN 978-0-87893-288-7 978-0-87893-289-4.

Jahn, A., Levitas, D., Holscher, E., Johnson, J. T., Sayal, A., jstaph, JohannesWiesner, Clucas, J., Tapera, T. M., and justbennet. Andrewjahn/AndysBrainBook:. Zenodo, January 2022.

Jenkinson, M., Beckmann, C. F., Behrens, T. E. J., Woolrich, M. W., and Smith, S. M. FSL. *NeuroImage*, 62(2):782–790, August 2012. ISSN 1053-8119. doi: 10.1016/j.neuroimage.2011.09.015.

Karras, T., Laine, S., and Aila, T. A Style-Based Generator Architecture for Generative Adversarial Networks, March 2019.

Kneeland, R., Jiang, W., Nunes, U. B., Lee, S. K., Scotti, P. S., Delorme, A., and Xu, J. ENIGMA: A Unified Lightweight EEG-to-Image Model for Multi-Subject Visual Decoding. In *NeurIPS 2025 Workshop on Foundation Models for the Brain and Body*, October 2025a.

Kneeland, R., Scotti, P. S., St-Yves, G., Breedlove, J., Kay, K., and Naselaris, T. NSD-Imagery: A benchmark dataset for extending fMRI vision decoding methods to mental imagery, June 2025b.

Krizhevsky, A., Sutskever, I., and Hinton, G. E. ImageNet Classification with Deep Convolutional Neural Networks. In *Advances in Neural Information Processing Systems*, volume 25. Curran Associates, Inc., 2012.

Lane, C., Kaplan, D. Z., Abraham, T. M., and Scotti, P. S. Scaling Vision Transformers for Functional MRI with Flat Maps, October 2025.

Lin, T.-Y., Maire, M., Belongie, S., Bourdev, L., Girshick, R., Hays, J., Perona, P., Ramanan, D., Zitnick, C. L., and Dollár, P. Microsoft COCO: Common Objects in Context, February 2015.

Mennen, A. C., Norman, K. A., and Turk-Browne, N. B. Attentional bias in depression, Understanding mechanisms to improve training and treatment. *Current opinion in psychology*, 29:266–273, October 2019. ISSN 2352-250X. doi: 10.1016/j.copsyc.2019.07.036.

Mennen, A. C., Turk-Browne, N. B., Wallace, G., Seok, D., Jaganjac, A., Stock, J., deBettencourt, M. T., Cohen, J. D., Norman, K. A., and Sheline, Y. I. Cloud-Based Functional Magnetic Resonance Imaging Neurofeedback to Reduce the Negative Attentional Bias in Depression: A Proof-of-Concept Study. *Biological Psychiatry. Cognitive Neuroscience and Neuroimaging*, 6(4):490–497, April 2021. ISSN 2451-9030. doi: 10.1016/j.bpsc.2020.10.006.

Mitchell, P., Lee, S. C. M., Yoo, P. E., Morokoff, A., Sharma, R. P., Williams, D. L., MacIsaac, C., Howard, M. E., Irving, L., Vrljic, I., Williams, C., Bush, S., Balabanski, A. H., Drummond, K. J., Desmond, P., Weber, D., Denison, T., Mathers, S., O'Brien, T. J., Mocco, J., Grayden, D. B., Liebeskind, D. S., Opie, N. L., Oxley, T. J., and Campbell, B. C. V. Assessment of Safety of a Fully Implanted Endovascular Brain-Computer Interface for Severe Paralysis in 4 Patients: The Stentrode With Thought-Controlled Digital Switch (SWITCH) Study. *JAMA Neurology*, 80(3):270–278, March 2023. ISSN 2168-6149. doi: 10.1001/jamaneurol.2022.4847.

Motiwala, A., Soldado-Magraner, J., Batista, A. P., Smith, M. A., and Yu, B. M. Brain–computer interfaces as a causal probe for scientific inquiry. *Trends in Cognitive Sciences*, 30(1):40–53, January 2026. ISSN 13646613. doi: 10.1016/j.tics.2025.06.017.

National Commission for the Protection of Human Subjects of Biomedical and Behavioral Research. The Belmont report: Ethical principles and guidelines for the protection of human subjects of research. https://www.hhs.gov/ohrp/regulations-and-policy/belmont-report/read-the-belmont-report/index.html, 1979.

Ozcelik, F. and VanRullen, R. Natural scene reconstruction from fMRI signals using generative latent diffusion, June 2023.

Ozcelik, F., Choksi, B., Mozafari, M., Reddy, L., and VanRullen, R. Reconstruction of Perceived Images from fMRI Patterns and Semantic Brain Exploration using Instance-Conditioned GANs, February 2022.

Pedregosa, F., Varoquaux, G., Gramfort, A., Michel, V., Thirion, B., Grisel, O., Blondel, M., Prettenhofer, P., Weiss, R., Dubourg, V., Vanderplas, J., Passos, A., Cournapeau, D., Brucher, M., Perrot, M., and Duchesnay, É. Scikit-learn: Machine Learning in Python. *Journal of Machine Learning Research*, 12(85):2825–2830, 2011. ISSN 1533-7928.

Pels, E. G., Aarnoutse, E. J., Leinders, S., Freudenburg, Z. V., Branco, M. P., van der Vijgh, B. H., Snijders, T. J., Denison, T., Vansteensel, M. J., and Ramsey, N. F. Stability of a chronic implanted brain-computer interface in late-stage amyotrophic lateral sclerosis. *Clinical Neurophysiology*, 130(10):1798–1803, October 2019. ISSN 1388-2457. doi: 10.1016/j.clinph.2019.07.020.

Peng, K., Wammes, J. D., Nguyen, A., Iordan, C. R., Norman, K. A., and Turk-Browne, N. B. Inducing representational change in the hippocampus through real-time neurofeedback. *Philosophical Transactions of the Royal Society B: Biological Sciences*, 379(1915):20230091, October 2024. ISSN 0962-8436. doi: 10.1098/rstb.2023.0091.

Podell, D., English, Z., Lacey, K., Blattmann, A., Dockhorn, T., Müller, J., Penna, J., and Rombach, R. SDXL: Improving Latent Diffusion Models for High-Resolution Image Synthesis, July 2023.

Prince, J. S., Charest, I., Kurzawski, J. W., Pyles, J. A., Tarr, M. J., and Kay, K. N. Improving the accuracy of single-trial fMRI response estimates using GLMsingle. *eLife*, 11:e77599, November 2022. ISSN 2050-084X. doi: 10.7554/eLife.77599.

Radford, A., Wu, J., Child, R., Luan, D., Amodei, D., and Sutskever, I. Language Models are Unsupervised Multitask Learners. 2019.

Radford, A., Kim, J. W., Hallacy, C., Ramesh, A., Goh, G., Agarwal, S., Sastry, G., Askell, A., Mishkin, P., Clark, J., Krueger, G., and Sutskever, I. Learning Transferable Visual Models From Natural Language Supervision, February 2021.

Ryoo, A. H.-W., Krishna, N. H., Mao, X., Azabou, M., Dyer, E. L., Perich, M. G., and Lajoie, G. Generalizable, real-time neural decoding with hybrid state-space models, November 2025.

Sauer, A., Lorenz, D., Blattmann, A., and Rombach, R. Adversarial Diffusion Distillation. In Leonardis, A., Ricci, E., Roth, S., Russakovsky, O., Sattler, T., and Varol, G. (eds.), *Computer Vision – ECCV 2024*, volume 15144, pp. 87–103. Springer Nature Switzerland, Cham, 2025. ISBN 978-3-031-73015-3 978-3-031-73016-0. doi: 10.1007/978-3-031-73016-0_6.

Scotti, P. S., Banerjee, A., Goode, J., Shabalin, S., Nguyen, A., Cohen, E., Dempster, A. J., Verlinde, N., Yundler, E., Weisberg, D., Norman, K. A., and Abraham, T. M. Reconstructing the Mind's Eye: fMRI-to-Image with Contrastive Learning and Diffusion Priors, October 2023.

Scotti, P. S., Tripathy, M., Villanueva, C. K. T., Kneeland, R., Chen, T., Narang, A., Santhirasegaran, C., Xu, J., Naselaris, T., Norman, K. A., and Abraham, T. M. MindEye2: Shared-Subject Models Enable fMRI-To-Image With 1 Hour of Data, June 2024.

Szegedy, C., Vanhoucke, V., Ioffe, S., Shlens, J., and Wojna, Z. Rethinking the Inception Architecture for Computer Vision. In *2016 IEEE Conference on Computer Vision and Pattern Recognition (CVPR)*, pp. 2818–2826, Las Vegas, NV, USA, June 2016. IEEE. ISBN 978-1-4673-8851-1. doi: 10.1109/CVPR.2016.308.

Tan, M. and Le, Q. V. EfficientNet: Rethinking Model Scaling for Convolutional Neural Networks, September 2020.

Tang, J., LeBel, A., Jain, S., and Huth, A. G. Semantic reconstruction of continuous language from noninvasive brain recordings. *Nature Neuroscience*, 26(5): 858–866, May 2023. ISSN 1546-1726. doi: 10.1038/s41593-023-01304-9.

Touvron, H., Lavril, T., Izacard, G., Martinet, X., Lachaux, M.-A., Lacroix, T., Rozière, B., Goyal, N., Hambro, E., Azhar, F., Rodriguez, A., Joulin, A., Grave, E., and Lample, G. LLaMA: Open and Efficient Foundation Language Models, February 2023.

Wallace, G., Polcyn, S., Brooks, P. P., Mennen, A., Zhao, K., Scotti, P. S., Michelmann, S., Li, K., Turk-Browne, N. B., Cohen, J. D., and Norman, K. A. RT-Cloud: A Cloud-based Software Framework to Simplify and Standardize Real-Time fMRI. *NeuroImage*, 257:119295, August 2022. ISSN 1053-8119. doi: 10.1016/j.neuroimage.2022.119295.

Wanjia, G., Favila, S. E., Kim, G., Molitor, R. J., and Kuhl, B. A. Abrupt hippocampal remapping signals resolution of memory interference. *Nature Communications*, 12(1): 4816, August 2021. ISSN 2041-1723. doi: 10.1038/s41467-021-25126-0.

Young, K. D., Siegle, G. J., Zotev, V., Phillips, R., Misaki, M., Yuan, H., Drevets, W. C., and Bodurka, J. Randomized Clinical Trial of Real-Time fMRI Amygdala Neurofeedback for Major Depressive Disorder: Effects on Symptoms and Autobiographical Memory Recall. *The American Journal of Psychiatry*, 174(8):748–755, August 2017. ISSN 1535-7228. doi: 10.1176/appi.ajp.2017.16060637.

Zhang, Y., Wang, Y., Azabou, M., Andre, A., Wang, Z., Lyu, H., Laboratory, T. I. B., Dyer, E., Paninski, L., and Hurwitz, C. Neural Encoding and Decoding at Scale, May 2025.

Zhou Wang, Bovik, A., Sheikh, H., and Simoncelli, E. Image quality assessment: From error visibility to structural similarity. *IEEE Transactions on Image Processing*, 13 (4):600–612, April 2004. ISSN 1057-7149, 1941-0042. doi: 10.1109/TIP.2003.819861.

# A. Appendix

## A.1. Fast Real-time (No Pretraining)

Table 3 shows reconstruction and retrieval evaluation metrics for the 3T real-time session without pretraining on NSD. We include a chance-level baseline corresponding to using randomly selected COCO images as the "reconstructions". Even without pretraining on NSD (i.e., only training the model with one session of 3T fMRI data), performance is still above the baseline, demonstrating that a large pretraining dataset (NSD) is not necessary for above-chance real-time decoding.

| Method | Latency | Low-Level | | | | High-Level | | | | Retrieval | |
|---|---|---|---|---|---|---|---|---|---|---|---|
| | | PixCorr ↑ | SSIM ↑ | Alex(2) ↑ | Alex(5) ↑ | Incep ↑ | CLIP ↑ | Eff ↓ | SwAV ↓ | Image ↑ | Brain ↑ |
| Fast real-time | 14.5s | 0.065 | 0.401 | 62.61% | 63.88% | 60.20% | 56.82% | 0.935 | 0.580 | 22% | 22% |
| Random Baseline | - | 0.014 | 0.277 | 50.25% | 50.99% | 50.37% | 50.38% | 0.982 | 0.655 | 1.6% | 1.4% |

*Table 3.* Latency and reconstruction/retrieval evaluation metrics on the 3T fast real-time pipeline without pretraining on NSD data. ↑ (↓) means higher (lower) scores are better. The baseline uses randomly chosen COCO images (excluding the "shared1000" images) as "reconstructions" for each test image.

## A.2. Replicating Real-time Pipelines using NSD

We used the same pretrained checkpoint as all other 3T and 7T experiments and then fine-tuned the model using the first session of fMRI data from NSD subj01. Differences in procedures include the number of training images per session (693 in 3T and 750 in NSD) and masking (the nsdgeneral mask was applied without an additional subject-specific reliability mask). The test trials consisted of the first presentation of the exact same subset of 50 special515 images used in the 3T data; however, these trials spanned multiple sessions rather than a single session. The results (Table 4, Figure 5, Figure 9) show a similar qualitative pattern of results to Table 1 where decoding performance improves with latency.

| Method | Latency | Low-Level | | | | High-Level | | | | Retrieval | |
|---|---|---|---|---|---|---|---|---|---|---|---|
| | | PixCorr ↑ | SSIM ↑ | Alex(2) ↑ | Alex(5) ↑ | Incep ↑ | CLIP ↑ | Eff ↓ | SwAV ↓ | Image ↑ | Brain ↑ |
| Offline | 1d | 0.228 | 0.330 | 84.5% | 93.1% | 85.5% | 78.5% | 0.832 | 0.448 | 78% | 82% |
| End-of-run real-time | 2.7m | 0.153 | 0.335 | 78.4% | 85.3% | 73.9% | 67.4% | 0.882 | 0.492 | 66% | 62% |
| Slow real-time | 36s | 0.174 | 0.333 | 77.2% | 82.6% | 69.9% | 65.2% | 0.897 | 0.505 | 58% | 58% |
| Fast real-time | 14.5s | 0.133 | 0.313 | 71.3% | 76.5% | 67.3% | 63.7% | 0.918 | 0.533 | 36% | 40% |

*Table 4.* Summary of latency and reconstruction/retrieval evaluation metrics for offline and real-time decoding pipelines using NSD subject 1. Reconstruction metrics are averaged over 5 random seeds; retrieval is deterministic. ↑ (↓) means higher (lower) scores are better.

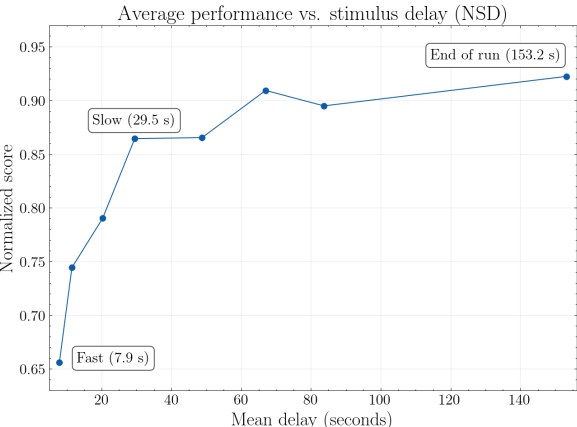

*Figure 5.* Hand-picked example reconstructions for different configurations on NSD subject 1.

### A.3. Replicating Stimulus Delay vs. Performance using NSD

We replicated the positive relationship between stimulus delay and performance (Figure 4) using NSD subj01 (Figure 6).

*Figure 6.* Average normalized performance across stimulus delays in NSD. Scores are min-max normalized per metric so that 0 corresponds to using random COCO images as reconstructions and 1 corresponds to offline NSD performance. Stimulus delay indicates the time elapsed after stimulus presentation before starting to analyze the neural response.

### A.4. Amount of Training Data vs. Performance in 3T and NSD

Figures 7 and 8 report normalized reconstruction and retrieval metrics as a function of training data. In both cases, models are pretrained on 7 out of 8 NSD subjects. For Figure 7, the model is fine-tuned on varying amounts of data from the 3T participant. For Figure 8, the model is fine-tuned on varying amounts of data from NSD subj01. Evaluation in both cases is performed on the same test set using the "fast real-time" pipeline. We observe that real-time performance improves with additional fine-tuning data in both datasets.

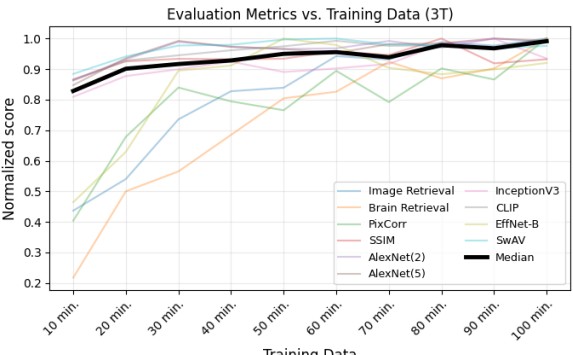

*Figure 7.* Evaluation metrics versus training data for the 3T subject. Scores are normalized to $[0, 1]$ per metric. The model is fine-tuned with varying amounts of data from the 3T subject (up to two sessions) and evaluated using the "fast real-time" pipeline.

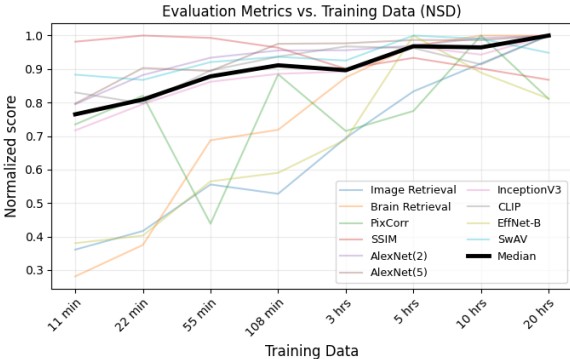

*Figure 8.* Evaluation metrics versus training data for NSD subj01. Scores are normalized to $[0, 1]$ per metric. The model is fine-tuned with varying amounts of data from subj01 and evaluated using the "fast real-time" pipeline.

## A.5. Additional 3T Data

Figure 9 is a comprehensive version of Figure 3 which aggregates metrics into low-level, high-level, and retrieval.

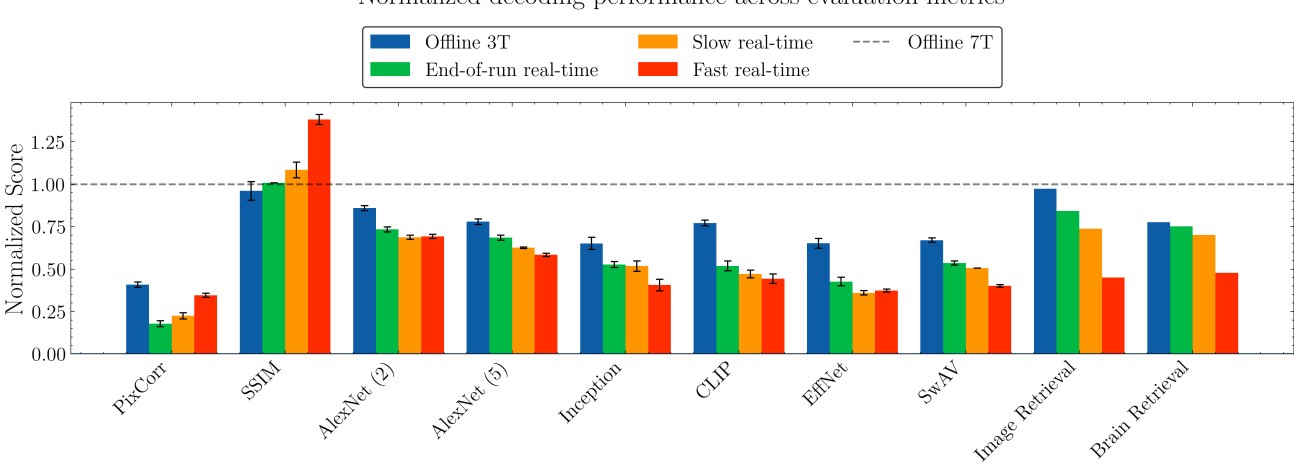

*Figure 9.* Normalized evaluation metrics for 3T offline and real-time pipelines. Scores are min-max normalized per metric so that 0 corresponds to using random COCO images as reconstructions and 1 corresponds to offline NSD performance. Bars show the mean and standard error across 5 random seeds.

Figure 10 contains additional, randomly chosen reconstructions as an extension of Figure 2.

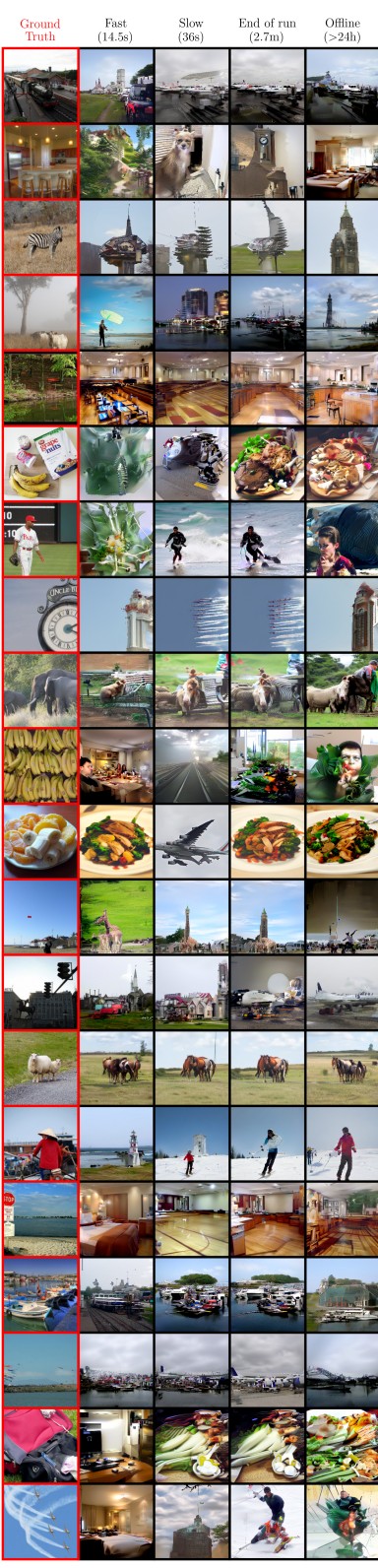

*Figure 10.* Randomly selected reconstructions for different configurations using 3T data.

### A.6. Experimenting with a Smaller Architecture

Due to the high dimensionality of the CLIP space used (∼426k dimensions), we considered the possibility that using a lower prediction dimensionality would be more appropriate given the limited amount of fine-tuning data and lower data quality as a result of the 3T scanner and real-time compatible preprocessing. As a result, we attempted to modify the architecture to use the last layer of OpenCLIP ViT-H/14 (Radford et al., 2021; Dosovitskiy et al., 2021), which contains just 1024 dimensions. We then generated reconstructions using SDXL Turbo (Sauer et al., 2025). We reasoned that while having a reduced dimensionality might limit the ceiling of expressivity of the model, this approach would enforce a stronger naturalistic prior on the reconstructed images and thus make the reconstructions appear more clear and visually appealing.

While the generated reconstructions from this architecture (Figure 11) indeed appeared more naturalistic and visually coherent as we intended, we found that, contrary to our intuition, the decoded contents were less faithful to the target images. This subjective assessment was supported by the quantitative metrics for reconstruction and retrieval (Table 5).

| Method | Latency | Low-Level | | | | High-Level | | | | Retrieval | |
|---|---|---|---|---|---|---|---|---|---|---|---|
| | | PixCorr ↑ | SSIM ↑ | Alex(2) ↑ | Alex(5) ↑ | Incep ↑ | CLIP ↑ | Eff ↓ | SwAV ↓ | Image ↑ | Brain ↑ |
| Offline 3T (avg. 3 reps.) | 1d | 0.082 | 0.360 | 67.3% | 76.6% | 64.4% | 71.2% | 0.908 | 0.572 | 14% | 22% |
| Offline 3T | 1d | 0.080 | 0.348 | 61.4% | 67.9% | 59.4% | 66.3% | 0.920 | 0.596 | 10% | 4% |
| End-of-run real-time | 2.7m | 0.054 | 0.352 | 61.6% | 68.5% | 60.0% | 61.6% | 0.942 | 0.604 | 12% | 2% |
| Slow real-time | 36s | 0.062 | 0.354 | 60.8% | 67.1% | 57.5% | 58.9% | 0.938 | 0.610 | 6% | 0% |
| Fast real-time | 14.5s | 0.042 | 0.346 | 61.0% | 61.5% | 57.5% | 57.8% | 0.952 | 0.632 | 12% | 4% |

*Table 5.* Latency and reconstruction/retrieval metrics for SDXL Turbo using 3T offline and real-time pipelines (reconstruction reported as the mean over 5 seeds; retrieval is deterministic). ↑ (↓) means higher (lower) scores are better.

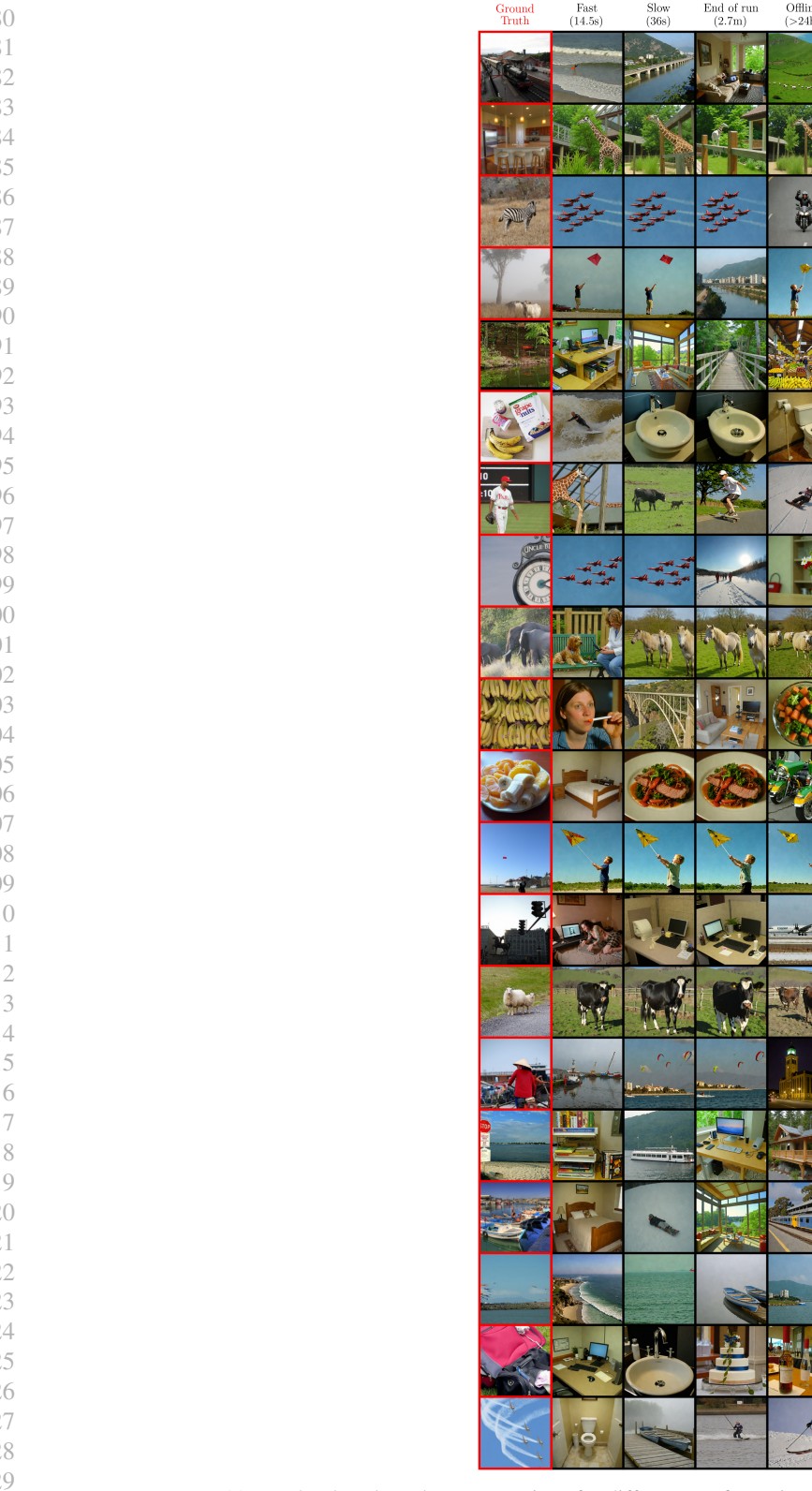

*Figure 11.* Randomly selected reconstructions for different configurations in the 3T subject, using SDXL Turbo.

### A.7. 3T MRI Acquisition

We acquired two spin-echo field map volumes (TR = 8000 msec, TE = 66 msec) in opposite phase encoding directions for fieldmap correction. We collected two whole-brain T1-weighted (T1w) MPRAGE images (one from each fMRI scan session; TR = 2300 msec, TE = 2.98 msec, voxel size = 1 mm isotropic, flip angle = 9°, 176 slices, Generalized Autocalibrating Partially Parallel Acquisitions [GRAPPA] acceleration factor = 2) as well as two T2-weighted turbo spin-echo (TSE) images (one from each session; TR = 11390 msec, TE = 90 msec, voxel size = 0.44 × 0.44 × 1.5 mm, flip angle = 150°, 54 slices acquired perpendicular to the long axis of the hippocampus, distance factor = 20%).

### A.8. Shifting Image Labels for Real-time

Prior to the start of the session, we assign an image label to each TR, shifted by ∼7.9 seconds post stimulus onset (for the "fast" real-time variation) to account for the BOLD response (Huettel et al., 2004). As a result, prior to analysis, each TR is assigned either a label of a previously seen image or is simply left blank. Varying the stimulus delay (e.g., between fast, slow, and end-of-run variations) therefore involves simply shifting these labels according to the desired amount of time.

## Glossary

**BOLD** Blood Oxygen Level-Dependent response. Rather than directly measuring electrical neural activity, fMRI measures changes in blood flow (primarily via the movement of deoxygenated hemoglobin) as an indirect proxy for neural activity. The temporally-extended BOLD response occurs on the order of seconds, peaking approximately 6-12 seconds after the underlying neural spikes (Huettel et al., 2004).

**Closed-loop Neurofeedback** An experimental paradigm in neuroimaging where a participant receives feedback based on a real-time readout of their own brain activity, with the goal of modulating this activity based on the experimenter's goals. Feedback typically uses reinforcement learning, usually providing visual feedback (such as an expanding circle) to indicate to the participant when their brain activity is moving in the desired direction.

**GLM** General Linear Model. A common step in fMRI data analysis that estimates the response strength of a particular brain voxel or region of interest to a single stimulus. The temporally-extended BOLD response unfolds over seconds; by modeling it with the HRF we can account for (deconvolve) the temporally-extended BOLD response, thereby converting a timeseries of measurements into a single response estimate per stimulus (Jahn et al., 2022).

**HRF** Hemodynamic Response Function. A function used to model the time course of the BOLD response that is triggered by neural activity. The HRF may vary as a function of subject, voxel, and stimulus, among other factors.

**Single-trial response estimates/betas** The regression beta coefficients per voxel obtained from the GLM. The GLM yields a single vector per stimulus, where the value of each element represents our estimate of each voxel or brain region's response to one stimulus. These betas (as opposed to the fMRI timeseries itself) are the inputs to the MindEye2 architecture.

**TR** Repetition Time. The sampling rate in fMRI, typically around 0.5-1 Hz. That is, a single 3-D volume of whole-brain functional activity is collected every 1-2 seconds.

**Voxel** Volumetric pixel. The basic neural unit in fMRI; voxels are typically 1-2 mm$^3$ isometric cubes that tile the brain uniformly. A voxel may contain several hundred thousand neurons (Huettel et al., 2004).

