# OpenReview forum: "Real-time Reconstruction of Human Visual Perception from fMRI"
_ICML.cc/2026/Conference — Submitted to ICML 2026_

### Official Review · Reviewer_7mj7 · 2026-03-06

**Soundness:** 3
**Presentation:** 3
**Significance:** 1
**Originality:** 2
**Overall Recommendation:** 2
**Confidence:** 4

**Summary:**

The authors implement a real-time image-decoding pipeline for fMRI recording, and evaluate the impact of various pre/post processing decision on image decoding.

**Compliance With Llm Reviewing Policy:**

Affirmed.

**Key Questions For Authors:**

1. I am unclear whether this is an actual *demonstration* (real time decoding did happen) or a proof-of-concept (real-time image generation did not happen, but could in principle be implemented).

2. I am not clear about the GLM step. Doesn’t this step require the recording of the full session? How is it fitted exactly (image presence, image classes?).

3. The performance of the offline 3T pipeline is close to offline 7T performance (Table 1, Figure 3).  -> This is quite surprising given the difference in SNR between 3t and 7t.

**Limitations:**

yes

**Strengths And Weaknesses:**

Strengths:
- This provides the tools (code, benchmark) to move forward on real-time fMRI-to-Image reconstruction
- Authors actually test it on a new 3T dataset, and obtain reasonable results (a rare effort, as many in the field are limited to existing datasets, which pose reproducibility issues)
- Paper is clearly written

Weakness:

- While the comparisons are technically interesting for anyone who wants to move towards real-time decoding, this paper is primarily an *implementation* of an existing real-time framework (RT Cloud) applied to an existing model (MindEye2). The actual ML contributions are somewhat limited, and makes me believe ICML may not be the right venue. There are a few ML experiments, e.g. finetuning on a new smaller dataset. While this is valuable, this kind of fine tuning on limited data has already been explored (e.g. Scotti et al 2024)

---

> ### Author Rebuttal · Authors · 2026-03-31
>
> We thank the reviewer for acknowledging our effort to pursue real-time applications and reproducibility. Below, we address the reviewer's concerns regarding originality and significance and Key Questions.
> ### Novelty
> We fully agree with the reviewer's assessment that our primary contribution centers on implementation. However, we view our integration of RT-Cloud with MindEye2 as a strength rather than a limitation.
>
> **Our work is the first to demonstrate that it is possible to reliably decode visual perception at a single-trial level with real-time fMRI.** This is significant because it narrows a clear gap between state-of-the-art offline fMRI decoding and the substantially simpler analyses that have typically been feasible in real-time settings, unlocking many experimental designs that were previously not possible. We are particularly excited about the possibility of providing closed-loop neurofeedback to modulate participants' visual cortical representations of individual stimuli, which we hope to pursue in future work.
>
> In addition, our implementation of real-time MindEye is open-sourced and officially supported by the RT-Cloud framework (links will be provided in the paper upon de-anonymization), providing an accessible foundation from which other researchers may investigate real-time deep learning methods. In this vein, we believe that this work is worth sharing with the community to demonstrate the frontier of machine learning-based analyses that are now possible with real-time fMRI. Further, we point to the ICML reviewer guidelines confirming that "originality may arise from creative combinations of existing ideas” or through an “application to a real-world use case", rather than requiring a completely novel decoding architecture or technique.
>
> ### Fine-tuning on limited data has been explored
> We agree with the reviewer that Scotti et al., 2024 have already explored fine-tuning on limited amounts of data. We believe that our results go beyond the MindEye2 explorations for two reasons: (1) We demonstrate that this approach transfers to a novel field strength with robustness to a different scanning protocol (different voxel size, TR, etc). (2) Fine-tuning on just one session is sufficient to enable real-time decoding in a new session with a significantly modified (real-time compatible) data processing pipeline.
>
> We do not claim that our use of limited training data is novel – especially given that our MindEye2 approach is based on the original paper which has already demonstrated this. However, prior to our work it was not clear to us whether this low data quantity would generalize to enable real-time decoding with a lower field strength and with real-time compatible preprocessing, both of which are significant departures from the MindEye2 paper. As a result, we believe that this finding is also of significant interest to the community.
>
> ### Key Questions
> 1. **This is an actual demonstration; real-time decoding did happen in the MRI scanner.** The simulated real-time analyses in this paper use data from another previously collected session. We refer the reviewer to this quote from the paper: "After developing these methods in simulation using previously collected 3T data, we successfully implemented this in a real-time fMRI session where we displayed reconstructions within seconds after the participant viewed the images. Here, we present simulated real-time analyses to test variations in preprocessing and analysis".
> 2. The GLM does not require data from the full session; it can simply be fit with as much data as is available at a given point in time. The GLM takes two primary inputs: the functional data (after preprocessing) and the design matrix, which specifies the time at which each image was shown. By convolving the design matrix with the hemodynamic response function (to account for fMRI's intrinsic hemodynamic delay), we fit a linear model to get a "beta" coefficient for each voxel. Specifically, we use a Least Squares Separate GLM (nilearn), which models the trial of interest "separate" from all other trials, resulting only in a beta for the target image (as opposed to all images simultaneously.
> **We have clarified this point in the Methods section of the revised paper and thank the reviewer for calling attention to this.**
> 3. We agree that it is striking that the decoding performance of the 3T and 7T subjects are close. This supports our claim that reliable real-time decoding is possible with 3T fMRI. We caution against interpreting the scores as directly comparing decoding accuracy in the two field strengths due to several differences including the participants, scan parameters, stimuli, etc. Future work may explore these factors more systematically, but we do agree with the reviewer that we expect 3T decoding to be worse (all else being equal) than 7T due to its lower intrinsic SNR.
> **We thank the reviewer for clarifying this and have updated Section 3.1 accordingly.**

---

### Official Review · Reviewer_GP1k · 2026-03-11

**Soundness:** 3
**Presentation:** 3
**Significance:** 3
**Originality:** 2
**Overall Recommendation:** 4
**Confidence:** 4

**Summary:**

The paper proposes a real-time pipeline for reconstructing perceived images from fMRI signals. The goal is to bridge the gap between state-of-the-art offline brain decoding methods and real-time neurofeedback systems. The authors integrate MindEye2 visual decoding model with a real-time fMRI processing framework.  The work demonstrates that visual reconstruction can be achieved in near real time. The system is evaluated using the NSD dataset for pretraining and fine-tuned with about one hour of data from a new participant. Overall, the paper presents a proof-of-concept system for real-time fMRI decoding highlighting its potential for applications such as closed-loop neurofeedback experiments.

**Compliance With Llm Reviewing Policy:**

Affirmed.

**Final Justification:**

I thank the authors for their detailed responses throughout the rebuttal period. The authors resolved most of my concerns. While I still believe that the method’s technical novelty and evaluation are somewhat limited, which the authors also acknowledge, I do find the overall contribution of the paper to be significant. The authors provide a practical tool for real-time fMRI and, in addition, offer by far the most detailed analysis of how different fMRI preprocessing choices influence decoding performance, which is an important and currently underexplored question. The existing analyses, together with the additional analyses the authors conducted during the rebuttal and promised to include in the revision, give this paper a unique and interesting contribution with meaningful potential impact.

Overall, I believe this paper provides a useful tool, interesting and potentially impactful analyses, and extensive discussion, and therefore should be accepted.

**Key Questions For Authors:**

Overall, the paper tackles an interesting and important task and moves in a needed direction of analysis. However, since the technical contribution itself is limited, I would expect the analysis to be deeper. For example, better examining the different factors and their effects, and showing (or at least explaining more clearly) the impact of this work on a broader range of experiments. Currently, the paper focuses mainly on image reconstruction, and it remains unclear how the findings generalize to other decoding or neurofeedback settings.

That said, I do believe the paper can make a valuable contribution to the field. I would be willing to increase my score if the authors strengthen the paper by adding deeper analysis of the main phenomena explored in the paper and by broadening the demonstrated impact, either through additional analysis or by evaluating the pipeline with more decoding models or experimental setups.

**Limitations:**

yes

**Strengths And Weaknesses:**

**Strengths:**
1. The paper tackles an interesting and impactful task of real-time fMRI decoding.
2. The analysis of how different fMRI preprocessing choices influence decoding performance is important and relatively underexplored.
3. The paper presents multiple analyses examining the influence of preprocessing choices on decoding performance.
4. The authors demonstrate promising transfer to a new subject on 3T machine using only one hour of data, which is encouraging for practical deployment.

---

**Weaknesses:**

1. **Technical contribution is limited.**
   Overall, the authors do not propose significant new methods to make the pipeline faster, and the work mainly combines existing technical tools. The paper also does not propose methodological improvements motivated by the presented findings and analyses (e.g., adaptations of the GLM or model architecture to better handle different temporal windows).

2. **Broader impact of the work should be clarified and better validated.**
   A main contribution suggested by the paper is that researchers may wish to select different points along the trade-off between real-time decoding speed and reconstruction accuracy. However, the paper mainly demonstrates that such a trade-off exists and analyzes it for image reconstruction, while the implications for other tasks remain unclear.  Even within decoding, the connection to practical neurofeedback pipelines is not well explored. For example, the trade-off might differ depending on whether the goal is recovering semantic information, low-level perceptual details, emotional content, or other attributes of the image. Exploring such distinctions would require evaluating models beyond full-image reconstruction.

3. **Stimulus delay explanation and analysis should be improved.**
   The authors observe that performance continues to improve as the delay increases to include data from the entire functional run, even though the BOLD response for a specific image should decay earlier. This phenomenon is interesting but insufficiently analyzed:

   **a.** Does this phenomenon occur only in the online setting, where preprocessing differs, or does it also appear in offline pipelines? Which specific processing component explains this effect?

   **b.** One explanation suggested is that longer delays allow capturing information about a voxel’s differential selectivity for an image. However, it is unclear why this would require data after the stimulus presentation. If most of the beta signal decays within ~20 seconds, is it important that the data come after the image presentation? Could similar information be obtained from earlier time points or even from other runs? Additional analysis would help clarify this point and may suggest an interesting improvement.

   **c.** The second explanation involving blank trials is unclear. As I understand it, blank trials are not directly tied to the GLM time window used for a given stimulus. It would be helpful to clarify how they support the proposed explanation.

4. **Robustness of the results across decoding models and subjects is not analyzed.**
   The experiments are conducted only with MindEye2 and on a single new 3T subject, which limits the strength of the conclusions. Evaluating the pipeline with additional decoding models or subjects would strengthen the analysis and its implications.

---

**Minor Comments:**

1. It would be helpful to provide more concrete examples of how image decoding from fMRI could be used in neurofeedback settings, and which properties of the decoded images would be most important for such applications.
2. The differences between offline preprocessing and real-time preprocessing and their impact on performance should be explained more clearly. It would also help to report more explicitly the time differences between the preprocessing variants.

---

> ### Author Rebuttal · Authors · 2026-03-31
>
> We thank the reviewer for their positive feedback on the impact of real-time decoding and on our explorations of real-time preprocessing. We share their excitement that our transfer learning results are encouraging for real-world deployment. Below, we address the reviewer’s concerns regarding our technical contributions, impact, and robustness of analysis.
> ### Technical contribution is limited
> > Overall, the authors do not propose significant new methods to make the pipeline faster, and the work mainly combines existing technical tools.
>
> We acknowledge the reviewer’s feedback on the originality of our work and refer them to the “Novelty” section of our rebuttal to Reviewer 7mj7 regarding these concerns.
>
> > The paper also does not propose methodological improvements motivated by the presented findings and analyses (e.g., adaptations of the GLM or model architecture to better handle different temporal windows).
>
> See below.
>
> ### Broader impact of the work should be clarified and better validated.
> We appreciate that the reviewer identified the importance of considering the trade-off between speed and accuracy, especially as similar deep-learning based approaches for real-time fMRI become feasible for other domains. We agree that it is important for researchers to consider this trade-off for their domain-specific applications, as decoding accuracy may underlie the efficacy of the neurofeedback intervention or real-time experimental design. While we are excited to see how similar real-time fMRI approaches can be implemented in the domains that the reviewer mentions (semantics, perception, emotion, etc), we believe that these explorations are beyond the scope of this paper. Our goal is to provide proof that it is possible to perform real-time, single-trial visual decoding; future work may explore other tasks and domains. For additional discussion, please see our response to Key Question 3 from Reviewer 5Bji, which refers to potential experimental designs related to other cognitive tasks mentioned by the reviewer.
> ### Stimulus delay explanation and analysis should be improved.
> We agree with the reviewer that this is an interesting phenomenon. We have made some preliminary attempts to investigate this effect; we tested the intuition that the additional data could come from earlier time points or runs. To explore this, we re-fit the real-time GLM while accumulating the design matrix and functional data across runs. We implemented this for the fast real-time setting and for now, our initial results are inconclusive: forward retrieval is higher than reported in the paper while backward retrieval is lower. However, it does seem to specifically benefit early trials in the run, which is consistent with the idea that the additional data is helpful in the case where there is otherwise very little available data from the run. We will validate this as soon as possible and additionally examine this effect for various stimulus delays. We are very interested in this promising analysis and we will include the results from these analyses in the revised version of the paper.
>
> The “blank trial” explanation of the stimulus delay effect is related to the fact that stimuli are presented in rapid succession (every 4 seconds) in this NSD-like design. This means that the BOLD responses to successive trials overlap. Including blank trials may improve beta estimates because they may introduce periods of time where the BOLD response can decay down to baseline, allowing the GLM to distinguish between the BOLD responses to consecutive images.
> ### Robustness of the results across decoding models and subjects is not analyzed.
> We agree that only including one new 3T subject for analysis is limiting. We point the reviewer towards Appendix A.2 and A.3, where we qualitatively replicate our core findings in NSD subj01. As mentioned in our response to Reviewer 5Bji (Key Question 2), we acknowledge that a limitation of our work is that we cannot make any claims about generalizability to other subjects, however we note that this problem is not unique to real-time decoding.
>
> In addition, we tested a simple baseline decoding model, described in our response to Key Question 1 from Reviewer 5Bji – this demonstrates above-chance retrieval accuracy using a very small model, showing an initial hint that real-time transfer learning may be doable with very simple architectures.
> ### Minor Comments
> 1. Please see our response to Reviewer 5Bji (Key Question 3) for concrete examples of neurofeedback applications.
> We will elaborate on the differences between the preprocessing pipelines in the paper.
>
> 2. The offline processing steps (fMRIPrep and GLMsingle) generally take several hours – on the order of 10-15 hours each – to run, compared with seconds to minutes for the real-time compatible pipeline.

---

> > ### Author Rebuttal · Reviewer_GP1k · 2026-04-01
> >
> > I thank the authors for their response to my concerns. The authors addressed some of my concerns, while some crucial ones are still unresolved and make me unsure whether to recommend acceptance of this paper. Generally, I see this paper as having two possible types of contribution: (i) a **technical contribution** of enabling better real-time decoding; and (ii) an **observational and analytical contribution** showing that transfer learning across scanners and single-trial real-time reconstruction are possible, and analyzing the requirement for real time fMRI decoding and the different importnat parameters, including the influence of fMRI preprocessing on decoding performance. In the current version, I feel that contribution (i) is very limited, as the technical contribtuion is low and very little was done to improve performance beyond a relatively simple combination of existing components. I do believe that contribution (ii) is important and relatively underexplored in the current literature; however, I feel that the analyses in this paper are underdeveloped and have little connection to, or explanation of, their relevance for real neurofeedback scenarios.
> >
> > I would be willing to change my score to accept if I were convinced on one of the following:
> >
> > a. **Show that the technical contribution is significant enough (Weakness 1 in my original review).**
> > While my concern was about the limited technical contribution, I think the authors’ response missed the point of my concern by referring instead to another reviewer’s novelty concern. While I do recognize the demonstration of single-trial real-time reconstruction, I still think that the technical contribution is limited. To clarify, this other contribution could possibly be achieved even without substantial methodological development, simply by applying MindEye2 to the new data. Since the authors present this as a pipeline that enables better single-trial reconstruction, rather than only as an observational paper, I would expect a stronger technical contribution. This could include new methodologies, analysis of different existing methodologies, suggestions for adapting current methods, etc. This is even more critical given that the contribution is also limited by the narrow evaluation setting, as acknowledged by the authors in their response to Weakness 4.
> >
> > b. **Explain the requirements of fMRI decoding for neurofeedback and provide better analysis (Weaknesses 2 and 3).**
> > What are the concrete scenarios in which real-time decoding is needed for neurofeedback? What aspects of the image are needed, and in which cases? What reconstruction quality would be sufficient for such use cases? These are crucial points for understanding the relevance and direction of this work. Moreover, why is decoding specifically interesting here? And what aspects of decoding matter most? Can we extract only semantic information? Only low-level information? For example, perhaps retrieval is sufficient for neurofeedback and full image reconstruction is unnecessary, in which case the pipeline should be designed differently. This clarification and discussion are needed in order to properly evaluate the contribution of this work. The authors did provide one example in their response to reviewer 5Bji, but without analysis or generalization to broader requirements. While I appreciate the initial experiments added in response to my third weakness, this aspect is still very underdeveloped. Better analysis of this point, as well as of the trade-offs and differing requirements for image reconstruction (e.g., is high-level information sufficient? Maybe only some metrics matter? How does this influence the time–quality tradeoff?), would significantly improve the paper.
> >
> > In its current state, I feel the paper is missing critical aspects needed to make it significant for the community, beyond being a technical tool that may be useful.

---

> > > ### Author Response · Authors · 2026-04-07
> > >
> > > We thank the reviewer for continuing to engage with our work and offering constructive feedback regarding the impact of our work.
> > >
> > > ## Technical Contribution
> > > The reviewer stated that this “could possibly be achieved even without substantial methodological development, simply by applying MindEye2 to the new data”. We respectfully disagree with this statement and believe that most previous work in the field of fMRI-to-image decoding (including MindEye2) relied on extensive preprocessing steps such as fMRIPrep and GLMsingle, taking several hours each.
> > >
> > > Given the reliance of prior work on such algorithms, it was unclear whether it was even possible to replace these carefully optimized pipelines and still retain any above-chance real-time decoding. We see our work as providing a valuable existence proof that fMRIPrep and GLMsingle are not necessary to support highly specific image decoding; we offer an alternative real-time compatible pipeline and further quantify the surprisingly small drop in performance in the transition from offline to real-time.
> > >
> > > With regard to RT-Cloud, this is the first demonstration that it (or any other real-time fMRI framework) can support the computational demands of a modern generative AI pipeline involving large neural networks such as MindEye2. In addition, we provide comprehensive documentation and open-source code which we believe will be a significant resource to the community.
> > > ## Neurofeedback
> > > We thank the reviewer for raising these questions. Perhaps the most fundamental question is: Can people learn from fMRI neurofeedback delivered at latencies corresponding to our “fast” and “slow” pipelines (10-30s)? Existing studies provide strong evidence that this level of delay is suitable, even reporting promising effects with 20-40s latencies [1,2]. To this point, [3] uses simulations to show that in certain situations, intermittent feedback may facilitate learning better than continuous feedback, especially under realistic assumptions regarding the blurred and delayed hemodynamic response of fMRI.
> > >
> > > Importantly, these studies delivered neurofeedback based on simple neural signals. Our framework can deliver much more fine-grained feedback within the same time budget. To give one example of how we plan to apply our pipeline in the coming year: An important question in memory research is how people learn to discriminate between very similar stimuli (e.g., learning to associate a photo of a barn with one person’s face, and a very similar photo of a barn with a different face). Prior theoretical work [4] suggests that participants’ success in this task depends on how well they represent the distinctive features of the barns, and predicts that successful discrimination occurs when distinctive features are represented strongly enough in visual cortex to evoke non-overlapping representations of the two barns in the hippocampus. To provide a causal test of this hypothesis, we plan to use our pipeline to reward participants when they attend to the unique features of the scene they are viewing. We will accomplish this by mapping fMRI data into CLIP space (“retrieval” part of the pipeline) and measuring if this is closer to the CLIP coordinate of the correct vs. the incorrect barn; the closer it is, the more reward participants will receive. We can then causally test whether inducing cortical distinctiveness repels hippocampal representations, leading to successful discrimination.
> > >
> > > An important feature of our pipeline is that it is modular; it is possible to only do retrieval at inference time. In the design above, this would save 4.6s as reconstruction could be skipped, bringing the latency down to 9.5s, as illustrated in the upper portion of Figure 1 in our manuscript (note that the MindEye2 authors showed that training on both objectives benefited the other). Prior feedback studies have used similar designs, with the limitation that they used a bespoke binary classifier (e.g., table vs. chair) [5]. A key benefit of our pipeline is that the model can be trained once (on a diverse set of scenes) and then applied to a large number of stimuli (e.g., discriminating between two barns, giraffes, mountains, etc), vastly expanding the amount of learning data that can be gathered from a single experimental session.
> > >
> > > **We thank the reviewer for clarifying these important points; we have revised the Discussion section accordingly.**
> > >
> > > [1] Neurofeedback Training Facilitates Awareness and Enhances Emotional Well-being Associated with Real-World Meditation Practice: A 7-T MRI Study
> > >
> > > [2] Upregulation of reward mesolimbic activity and immune response to vaccination: A randomized controlled trial
> > >
> > > [3] Self-regulation strategy, feedback timing and hemodynamic properties modulate learning in a simulated fMRI neurofeedback environment
> > >
> > > [4] A neural network model of differentiation and integration of competing memories
> > >
> > > [5] Inducing representational change in the hippocampus through real-time neurofeedback

---

### Official Review · Reviewer_5Bji · 2026-03-12

**Soundness:** 3
**Presentation:** 3
**Significance:** 2
**Originality:** 2
**Overall Recommendation:** 3
**Confidence:** 4

**Summary:**

This paper adapts RT-Cloud and a streamlined MindEye2 pipeline to make single-trial fMRI-to-image decoding feasible in a real-time-compatible 3T setting, and studies the trade-off between decoding quality and latency.

**Compliance With Llm Reviewing Policy:**

Affirmed.

**Final Justification:**

Overall, the rebuttal improves the positioning of the paper and partially resolves my questions about baselines and practical scope, but it does not sufficiently change my view on originality or broader validation. I therefore keep my overall recommendation at Weak Reject.

**Key Questions For Authors:**

1.	Could the authors include a simpler real-time baseline under the same latency budget, such as a retrieval-only or lighter projection model, to clarify how much of the gain comes from the adapted MindEye2 stack versus the real-time pipeline itself?
2.	How should readers think about cross-subject and cross-site generalization, given that the main 3T demonstration uses a single author-participant? Even a clearer discussion of expected failure modes would help.
3.	Since the fast setting is around 15 seconds and the paper finds an elbow around 30 seconds, which concrete closed-loop applications do the authors believe are already realistic here, and which still seem out of reach?

**Limitations:**

The paper discusses some practical limitations of fMRI and notes the leakage issue in pretraining, but the manuscript would benefit from a more explicit statement that the current evidence is best viewed as a feasibility demonstration, not yet a broadly validated real-time decoding solution.

**Strengths And Weaknesses:**

**Strengths:**
- From an engineering perspective, the paper does a solid job of addressing the deployment challenges of computationally intensive fMRI reconstruction models such as MindEye2.

- The paper presents a real-time fMRI-to-image decoding pipeline and discusses the associated offline and real-time preprocessing choices in a fairly comprehensive way.


**Weaknesses:**

1.	My main concern is novelty. The paper is mostly an engineering adaptation and integration of existing components, including RT-Cloud, a condensed MindEye2 pipeline, and established preprocessing steps, rather than a new decoding method or a new learning idea. For ICML, that makes the technical contribution feel somewhat limited.
2.	Relatedly, the evaluation is centered on one adapted architecture. It is still unclear whether the proposed real-time pipeline generalizes well to other decoding backbones, or whether a simpler baseline under the same latency budget would already recover a large fraction of the reported performance.
3.	I also think the scope of the “real-time” claim should be stated a bit more carefully. The fast setting is about 14.8 seconds after stimulus onset, and the paper itself suggests that a better trade-off may occur around 30 seconds. That is still interesting and potentially useful, but it narrows the set of applications where the method would function as genuinely real-time feedback.

---

> ### Author Rebuttal · Authors · 2026-03-31
>
> We thank the reviewer for their constructive feedback and are grateful for their acknowledgement of our efforts addressing real-world deployment and “comprehensive” comparisons of preprocessing pipelines. Below, we address the reviewer’s concerns regarding novelty, baseline, generalizability, Key Questions, and limitations.
> ### Novelty
> We acknowledge the reviewer’s feedback on the originality of our work and refer them to the “Novelty” section of our rebuttal to Reviewer 7mj7 regarding these concerns.
> ### Alternate architectures
> See rebuttal to Key Question 1.
> ### Real-world applications
> We agree with the reviewer that it is important for real-time feedback applications to consider the desired time frame. However, rather than categorizing the set of applications as “real-time” or not, we consider the cognitive process of interest when considering how fast a given analysis needs to be. Slower cognitive processes that unfold over longer timescales may be able to benefit from slower neurofeedback, while the fastest analyses will toe the line between speed and accuracy. This is analogous to the choice of voxel size and TR length when designing an fMRI experiment, where the choice of such parameters will affect the data resolution but also signal-to-noise ratio. See the response to Key Question 3 below for additional discussion.
> ### Key Questions
> 1. We thank the reviewer for the suggestion to include a simple baseline model. We trained a small 9M parameter MLP (2048 hidden size) using MSE loss to predict CLIP embeddings (1664 dimension; pooled over the sequence dimension) from the training session, then evaluated on the held-out session with real-time processing. This achieves ~6% retrieval accuracy with a candidate pool size of 50 (1/50 = 2%). This suggests that it is possible to reach above-chance real-time retrieval with a simple model, but that there is substantial gain with larger models. **We will include a comprehensive baseline evaluation with all reconstruction and retrieval metrics in the revised paper.**
>
> 2. We agree with the reviewer that a limitation of our work is sample size. This is partially addressed by Appendix A.2 and A.3, where we replicate our main results using NSD subj01. That being said, we acknowledge that we cannot make claims about the generalizability to other subjects or sites. Although we believe there is no reason to think that our real-time approach would not work in other scanning sites, we agree that it is important to test these claims empirically in future work. Regarding failure modes, we expect that the more different the training and testing sessions are, the worse transfer one will experience. For example, some important factors to consider include the duration between scans, stimulus distribution overlap, and task similarity.
>
> 3. We appreciate that the reviewer is interested in concrete applications. We believe that in principle any analysis that is possible to do offline should also be doable in real-time, with a performance drop. Importantly, the window for closed-loop feedback depends on the time course of the cognitive process.
>
>     For example, a previous closed-loop neurofeedback study based on visual responses [1] used a 16-second stimulus presentation which is quite close to our reported “fast” real-time latency. We believe that this is a reasonable latency at which to perform neurofeedback, since participants can take time to focus on a particular mental state and think about the item being presented within this time frame. An example design that could use a 30-second delay could be a study instructing participants to immerse themselves in memories and emotions associated with an image. In this case, aiming for a shorter feedback loop may not even be desirable because it would be incompatible with the process of interest. **We thank the reviewer for raising these topics; we have added a discussion of these topics in the revised paper.**
>
>     Finally, an example of a task that is still out of reach is mental imagery decoding. It has yet to be shown that it is possible to reliably decode imagined visual perception from single-trial fMRI, regardless of the available processing time. As such, there is no reason to expect that real-time mental imagery decoding is now possible. If and when mental imagery decoding becomes possible, a similar RT-Cloud-based framework could be used to adapt this analysis to real-time fMRI.
> ### Limitations
> We agree with the reviewer’s point that we view the current work as a demonstration of feasibility rather than a large-scale validation of real-time decoding across subjects and stimulus sets. **We have added a Limitations section to the paper’s end-matter where we explicitly discuss this and the other points raised in Key Question 3.** We appreciate the reviewer’s focus on distinguishing what is and isn’t currently possible with this technology.
>
> [1] Sculpting new visual categories into the human brain. Iordan et al.

---

> > ### Author Rebuttal · Reviewer_5Bji · 2026-04-04
> >
> > Thank you for the rebuttal. The response addresses part of my concerns. In particular, the added simple MLP baseline is helpful: it shows that a lightweight real-time model can achieve above-chance retrieval, while still leaving a substantial gap to the adapted MindEye2 pipeline. I also appreciate the clearer discussion of application timescales and the explicit clarification that the present work should be viewed as a feasibility demonstration rather than a broadly validated real-time decoding solution.
> >
> > That said, my main concern about novelty remains. I still view the paper primarily as an engineering adaptation and integration of existing components, rather than a new decoding method or a new learning idea. The additional baseline strengthens the empirical case, but it does not materially change my assessment that the technical contribution is limited for ICML. In addition, the main 3T demonstration is still centered on a single author-participant, so broader claims about cross-subject and cross-site generalization remain premature.

---

> > > ### Author Response · Authors · 2026-04-07
> > >
> > > We thank the reviewer for their continued feedback regarding novelty and agree that this is essential to clarify. We point the reviewer to our "Reply Rebuttal Comment" to Reviewer GP1k, where we provide clarifications about technical contributions as well as novel use cases that are made possible by our work.
> > >
> > > Notably, we believe that our comments address the reviewer's original concerns about novelty:
> > > > My main concern is novelty. The paper is mostly an engineering adaptation and integration of existing components, including RT-Cloud, a condensed MindEye2 pipeline, and established preprocessing steps, rather than a new decoding method or a new learning idea. For ICML, that makes the technical contribution feel somewhat limited.
> > >
> > > As mentioned in our other response, we believe that we have made non-trivial changes to the preprocessing pipeline that represent a significant departure from most other work in the field and additionally quantify the drop in performance in the transition to real-time.
> > >
> > > Finally, we believe that the "Neurofeedback" section of our aforementioned response to Reviewer GP1k addresses the reviewer's original questions regarding practical applications of our pipeline.
> > >
> > > We hope that these clarifications address the reviewer's concern about novelty. In sum, we believe that while our work indeed is an integration of existing components, this integration is non-trivial due to the preprocessing differences from prior work, documentation of various parts of the preprocessing and analysis pipeline, as well as the novel and exciting experimental designs that are made possible by this work -- which we and (we hope) others will pursue in the near future.

---

### Official Review · Reviewer_ShuP · 2026-03-12

**Soundness:** 3
**Presentation:** 3
**Significance:** 3
**Originality:** 3
**Overall Recommendation:** 4
**Confidence:** 4

**Summary:**

This work presents the first end-to-end real time fMRI to Image reconstruction and retrieval pipeline. The work adapts MindEye2 to operate within the compute constraints of real-time analysis using the software RT-Cloud. The model is pretrained on 7T NSD subjects and then fine-tuned on 1 hour of 3T data from a new participant, before the model is ready for real time decoding in a subsequent session within 15-40 seconds of image onset. They combine online motion correction, real time GLM convolution and fast reconstruction. The pipeline is evaluated on a newly collected dataset and on the left out NSD subject.

**Compliance With Llm Reviewing Policy:**

Affirmed.

**Final Justification:**

The paper presents a meaningful systems contribution by demonstrating a real-time fMRI-to-image decoding pipeline under practical constraints, which is a valuable step toward deploying such methods in real-world settings. However, the work relies heavily on an adapted MindEye2 architecture and provides limited evaluation beyond the specific dataset and setup, making it difficult to assess generality. In particular, broader validation across datasets or decoding approaches, as well as a deeper analysis of the preprocessing choices relative to standard offline pipelines, would strengthen the contribution.

**Key Questions For Authors:**

1. How many trials in the real-time session produced valid reconstructions/retrievals within the stated latency? If the authors could provide per-trial timestamps and failure rates it would maybe help.

2. What hardware (GPU model(s), CPU, storage, network bandwidth/latency) powered the RT-Cloud deployment, and how sensitive are the timing results to less capable hardware or on-prem clusters?

3. Can you report retrieval performance with larger candidate pools to better assess scalability? How does top-k change as pool size grows?

**Limitations:**

yes

**Strengths And Weaknesses:**

**Strengths:**
1. Technical novelty: Adapts a SoTA brain decoding pipeline to real-time constraints and is the first (to the best of my knowledge) to demonstrate end-to-end real time decoding feasibility on commodity 3T fMRI

2. Evaluation is mostly thorough with multiple real-time compatible settings that trade latency for accuracy and has comparisons to stronger offline baselines (fMRIPrep + GLMSingle)

3. Real-time single-trial visual reconstruction from non-invasive human fMRI fills a notable gap between sophisticated offline decoders and simpler real-time neurofeedback tools.

**Weaknesses:**

1. The retrieval pool is small (50 images), which inflates top-1 accuracy relative to more realistic candidate set sizes. I would recommend the authors experiment with open-set retrieval to estimate scalability

2. If I understand correctly, the real-time GLM uses a canonical HRF and does not adapt the HRF variability across brain regions or over time, potentially capping single-trial accuracy. Can you comment on how robust this is to inter-session variability and motion beyond just affine corrections?

3. The offline NSD value is surprisingly low, even with the low level and refined reconstruction step removed. Have you tried other pipelines that use simpler CLIP spaces or have advanced cross subject alignment techniques to see if your real time reconstruction can be improved beyond current performance. For example, even a simple alignment technique like the one used by [1] could maybe turn out to be better than a MindEye2 pipeline missing major components.

4. The evaluation uses images drawn from the NSD stimulus set, which is also the dataset used for model pretraining. This raises the possibility that performance may partially benefit from stimulus distribution similarity. How would performance be affected if the subject were to view images from a different subset eg: THINGS 3T fMRI instead. I get that its not possible to run real time experiments with it, but maybe an offline evaluation of THINGS 3T or BOLD5000 would serve as a decent benchmark to evaluate how performance might be affected if users were to conduct real time tests on a different set of images.


[1] Through their eyes: multi-subject Brain Decoding with simple alignment techniques. Matteo Ferrante et al.


I am leaning toward a borderline accept. The paper addresses an important practical gap in brain decoding by demonstrating a real-time reconstruction pipeline. However, a significant portion of the contribution appears to lie in systems integration and engineering rather than in new decoding methodologies. While this systems contribution is valuable, it makes the fit with the main research track of a machine learning conference somewhat less clear.

---

> ### Author Rebuttal · Authors · 2026-03-31
>
> We thank the reviewer for acknowledging the novelty of our contribution and the “notable gap” that we are aiming to close between single-trial decoding and existing real-time tools. We also appreciate their comment that we are addressing an “important practical” problem. Below, we address their concerns regarding retrieval pool size, preprocessing and architectural choices, image distribution, and Key Questions.
> ### Retrieval pool
> We appreciate the reviewer’s suggestion to expand the retrieval pool size and agree that retrieval from a large image pool would be a good measure of scalability. We will compute top-1 retrieval accuracy across varying pool sizes from size two to the maximum number of test images in the session and include these results in the revised paper.
> ### HRF variability and preprocessing
> We appreciate the reviewer discussing HRF variability and agree that this is an interesting direction. We explored this idea in two ways: (1) We used real-time compatible preprocessing followed by GLMsingle (which determines the optimal HRF per voxel). This did not improve decoding accuracy over the standard (nilearn) GLM with a canonical HRF. (2) We used real-time compatible preprocessing followed by the standard GLM, but using HRFs selected by GLMsingle from the training session. Unfortunately, this also did not improve decoding accuracy.
>
> One possible interpretation of these null results is that HRF variability may be secondary for real-time decoding performance compared to preprocessing quality or cross-session alignment/transfer. **We thank the reviewer for this interesting idea and have added these explorations to the revised paper.**
>
> We believe that our approach is robust to some level of inter-session variability, since we perform real-time decoding on a new session. However, we do not make claims about the impact of this variability on decoding performance due to our limited data. We did not examine motion-related confounds systematically, though we expect that instances of significant head motion would impede decoding (as with offline analyses). Future pipelines may explore additional preprocessing steps such as volume censoring [1].
> ### Alternate architectures
> We appreciate that the reviewer suggests smaller CLIP spaces or alignment approaches (such as hyperalignment in the referenced paper); we have also been exploring similar ideas. Following the first idea, we refer to Appendix A.6 and Table 5, where we tested the idea of predicting a smaller CLIP space (which would offer a stronger naturalistic prior, and thus potentially would be more suitable for the lower data quality of real-time processing). Unfortunately, the decoding performance was worse.
>
> Regarding alignment approaches, we attempted to hyperalign between the train and test session, but this did not improve decoding performance. However, this was not informed by the referenced paper; we have yet to explore cross-subject alignment. We are actively exploring alternative decoding approaches; more accurate decoders will only improve downstream scientific and clinical applications.
> ### Image distribution mismatch
> We completely agree that the shared image distribution between train and test likely has a significant impact on decoding. Unfortunately, we do not have access to fMRI responses to THINGS or BOLD5000 images (which would require additional scanning sessions), so we are unable to address this question with our current data. Related topics are discussed by Shirakawa and colleagues [2]. We hope to pursue future work that moves beyond the NSD stimuli.
> ### Key Questions
> 1. We assume the reviewer is asking whether any computational glitches occurred during the real-time session. Every trial was analyzed within the expected latency (summarized in Table 2) and a valid output was computed each time.
> 2. RT-Cloud is modular and can be run on many systems. We have successfully tested our pipeline on several Linux setups with single GPUs. Development was primarily performed on Linux with a NVIDIA RTX 6000 Ada Generation GPU (49GB VRAM) located in the MRI control room. Roughly 32GB of VRAM is used at runtime. We acknowledge that most MRI scanners do not have local GPU workstations; in these cases, users can use RT-Cloud’s cloud capabilities, removing the need to install costly local setups. As long as the network connection is reliable, this should not impact latency. **We thank the reviewer for bringing up these important considerations and we have added some discussions on this topic to the revised paper.**
> 3. See response above.
>
> Finally, we acknowledge the reviewer’s concerns around the fit of our work to a machine learning venue and refer them to the “Novelty” section of our rebuttal to Reviewer 7mj7 regarding these concerns.
>
> [1] Statistical improvements in functional magnetic resonance imaging analyses produced by censoring high-motion data points. Siegel et al.
>
> [2] Spurious reconstruction from brain activity. Shirakawa et al.

---

> > ### Author Rebuttal · Reviewer_ShuP · 2026-04-03
> >
> > Thanks for the response. My individual questions regarding the paper have been addressed. I see the other reviewers also have concerns about the fit of the work to the venue and the dependency on MindEye2/lack of comparisons. Since my score is already positive I will maintain my current score.

---

> > > ### Author Response · Authors · 2026-04-07
> > >
> > > We thank the reviewer for their overall positive comments.
> > >
> > > Regarding the reviewer's original question about scalability, here are the top-1 retrieval accuracy scores for varying pool sizes for the "fast" and "end-of-run" variations. The numbers reported here are the mean ± stdev over 200 random draws of candidate images for each pool size. As expected, there is a monotonic decrease in top-1 retrieval as the pool size increases, moderated by the latency, which seems to protect this decline in accuracy as pool size grows. The intermediate latencies fall between these scores. We will include the full results in the revised version of the paper but omit the full range of intermediate latencies here for brevity.
> > >
> > > | Pool size | Fast real-time | End-of-run real-time |
> > > |-----------|----------------|----------------------|
> > > | 2         | 89.6±0.3       | 92.2±0.2             |
> > > | 5         | 72.7±0.3       | 83.5±0.2             |
> > > | 10        | 59.5±0.3       | 77.3±0.3             |
> > > | 20        | 50.2±0.3       | 71.2±0.2             |
> > > | 30        | 45.6±0.2       | 67.2±0.3             |
> > > | 40        | 41.9±0.2       | 64.8±0.2             |
> > > | 50        | 40.2±0.2       | 62.9±0.2             |
> > > | 70        | 36.9±0.2       | 60.0±0.2             |
> > > | 100       | 34.4±0.2       | 56.5±0.2             |
> > > | 150       | 31.1±0.2       | 53.4±0.2             |
> > > | 200       | 28.5±0.2       | 50.9±0.2             |
> > > | 300       | 25.3±0.2       | 47.3±0.2             |
> > > | 400       | 22.4±0.1       | 44.5±0.1             |
> > > | 531       | 20.0±0.0       | 42.0±0.0             |

---

### Decision · Program_Chairs · 2026-04-30

**Decision:**

Reject

**Comment:**

Real-time fMRI decoding is in generally challenging. Levarging RT-Cloud, an open-source, scalable cloud-based platform for real-time fMRI, this study implements an algorithm for reconstructing perceived natural images (MindEye2) with real-time fMRI. There is some discrepancy among reviewers. Several reviewers expressed concerns on its theoretical novelty and the limited evaluation. The disadvantages outweigh the advantages. I tend to reject this paper.